# Genome-wide mega-analysis identifies 16 loci and highlights diverse biological mechanisms in the common epilepsies

The International League Against Epilepsy Consortium on Complex Epilepsies

The epilepsies affect around 65 million people worldwide and have a substantial missing heritability component. We report a genome-wide mega-analysis involving 15,212 individuals with epilepsy and 29,677 controls, which reveals 16 genome-wide significant loci, of which 11 are novel. Using various prioritization criteria, we pinpoint the 21 most likely epilepsy genes at these loci, with the majority in genetic generalized epilepsies. These genes have diverse biological functions, including coding for ion-channel subunits, transcription factors and a vitamin-B6 metabolism enzyme. Converging evidence shows that the common variants associated with epilepsy play a role in epigenetic regulation of gene expression in the brain. The results show an enrichment for monogenic epilepsy genes as well as known targets of antiepileptic drugs. Using SNP-based heritability analyses we disentangle both the unique and overlapping genetic basis to seven different epilepsy subtypes. Together, these findings provide leads for epilepsy therapies based on underlying pathophysiology.

A full list of consortium members appears at the end of the paper. Correspondence should be addressed to S.F.B. (email: s.berkovic@unimelb.edu.au), B.P.C.K. (email: B.P.C.Koeleman@umcutrecht.nl) or G.L.C. (email: gcavalleri@rcsi.ie)

The epilepsies are a group of brain disorders characterized by recurrent unprovoked seizures affecting up to 65 million people worldwide[1]. There are many different types of epilepsy, and its classification has recently evolved, driven by advances in clinical phenotyping, imaging, and genetics[2]. Since the identification of *CHRNA4* as a cause of autosomal dominant nocturnal frontal lobe epilepsy[3], genes underlying many different rare monogenic forms of epilepsy have been characterized, and discovery in this area has accelerated with the application of next generation sequencing[4]. This is particularly true of the relatively rare but devastating infantile group of epileptic encephalopathies, which are now emerging as a genetically heterogeneous group of largely de novo dominant disorders[5]. In contrast, single gene causes of the more common forms of epilepsy appear to be relatively rare. The common forms broadly comprise generalized and focal epilepsies, with the former having the highest heritability, yet the lesser yield in single gene discovery[6]. These common forms are likely multifactorial, with a significant and complex genetic architecture[7–9].

Consistent with the experience from many other disease fields, early attempts to disentangle the genetic architecture of the more common, sporadic forms of epilepsy were limited by study power and scope[10–14]. In 2011, the International League Against Epilepsy (ILAE) launched the Consortium on Complex Epilepsies, to facilitate meta-analysis in epilepsy genomics. In 2014, the first such meta-analysis was reported comprising 8696 cases and 26,157 controls. This led to the identification of 2q24.3, 4p15.1, and 2p16.1 as epilepsy loci[15].

Here we present an expanded analysis involving 15,212 cases and 29,677 controls, which leads to identification of 16 genome-wide significant loci. Importantly, 11 of these loci are associated with the genetic generalized epilepsies; the group of epilepsies where despite having the highest heritability we have made the least genetic progress to date. We show that the genes associated with each locus are biologically plausible candidates, despite having diverse functions, particularly as there is a significant enrichment for known monogenic epilepsy genes and anti-epileptic drug targets.

## Results

**Study overview**. We performed a genome-wide mega-analysis on the ILAE Consortium cohort now comprising 15,212 epilepsy cases, stratified into 3 broad and 7 subtypes of epilepsy, and 29,677 control subjects (Supplementary Table 1). The current study includes a further 6516 cases and 3460 controls in addition to the 8696 cases and 26,157 controls from our previously published analysis[15]. Thus, this mega-analysis is not a formal replication of our previously published meta-analysis. We do not attempt any formal replication of novel association signals detected in this analysis. Furthermore, 531 cases of Asian descent, and 147 cases of African descent were included through a meta-analysis. However, we refer to our GWAS as a mega-analysis as the vast majority of our samples (96%) were analyzed under that framework.

At the broadest level, cases were classified as (a) focal epilepsy where seizures arise in a restricted part of the brain, a form traditionally not regarded as genetic although a number of genes for monogenic forms have been identified; (b) genetic generalized epilepsy where seizures arise in bilateral networks and evidence for a genetic component is very strong, yet genes have been hard to identify, and (c) unclassified epilepsy[2,16].

Subjects were assigned to three broad ancestry groups (Caucasian, Asian and African-American) according to results of genotype-based principal component analysis (Supplementary Fig. 1). Linear-mixed model analyses were performed stratified by ethnicity and epilepsy subtype or syndrome, after which trans-ethnic meta-analyses were undertaken.

**Genome-wide associations**. Our analysis of all epilepsy cases combined revealed one novel genome-wide significant locus at 16q12.1 and reinforced two previous associations at 2p16.1 and 2q24.3 (Fig. 1 and Supplementary Fig. 2)[15]. When conditioning on the top SNP within the 2q24.3 locus, we demonstrate the existence of a second, independent signal within that locus (Supplementary Fig. 3). This locus was also significantly associated with focal epilepsy. Our analysis of genetic generalized epilepsy uncovered 11 genome-wide significant loci, of which seven are novel (Fig. 2).

Considering that focal and generalized epilepsy are clinically broad and heterogeneous classifications, we next assessed whether loci are specifically associated with any of the seven most common focal epilepsy phenotypes and genetic generalized epilepsy syndromes (Supplementary Fig. 4 and 5). We found a novel genome-wide significant association with juvenile myoclonic epilepsy (JME) and two novel loci associated with focal epilepsy with hippocampal sclerosis. Moreover, we found two genome-wide significant associations with childhood absence epilepsy (CAE) in loci that were previously associated with absence epilepsy and generalized epilepsy[12]. We did not find any significant loci associated with generalized epilepsy with tonic-clonic seizures (GTCS) alone, juvenile absence epilepsy (JAE), lesion-negative or lesional focal epilepsy (other than hippocampal sclerosis). Further analysis of the association signals for each locus in the different syndromes suggested that some signals display specificity for a single subtype, while others show evidence for pleiotropy (Supplementary Fig. 6). However, the relatively small sample sizes of these phenotype subsets warrant caution for over-interpretation.

In total, we found 11 novel genome-wide significant loci associated with the epilepsies and we replicated the association of five previous known loci[12,15] (Supplementary Fig. 7). Two previous reports of association did not reach our threshold for significance. This included a locus (rs2292096; 1q32.1) for focal epilepsy detected in an Asian population[14] ($p = 0.057$ in our trans-ethnic fixed-effects meta-analysis), and rs12059546 (1q43) detected previously for JME[12] ($p = 7.4 \times 10^{-5}$ in our Caucasian-only BOLT-LMM analysis).

**Gene mapping and biological prioritization**. The genome-wide significant loci from all analyses were mapped to a total of 146 genes (Supplementary Data 1) using a combination of positional mapping (±250 kb from locus) and significant distal 3D chromatin interactions of the locus with a gene promoter (FDR < $10^{-6}$). Considering that most loci encompass several genes, we devised criteria to systematically prioritize the most likely candidate genes per locus based on established bioinformatics methods and resources. This biological prioritization was based on six criteria (Fig. 2), similar to previous studies[17,18]. Each gene was given a score based on the number of criteria that were met (range 0–6). The gene(s) with the highest score in each locus, with a minimum of 2, were defined as biological epilepsy risk genes. We validated this approach using established epilepsy genes within our data (Supplementary Table 2). Using this approach, we were able to refine these loci to the 21 most likely biological epilepsy genes (Fig. 2).

These prioritized genes include seven ion-channel genes (*SCN1A, SCN2A, SCN3A, GABRA2, KCNN2, KCNAB1,* and *GRIK1*), three transcription factors (*ZEB2, STAT4* and *BCL11A*), the histone modification gene *BRD7*, the synaptic transmission gene *STX1B* and the pyridoxine metabolism gene *PNPO*. Notably, a conditional

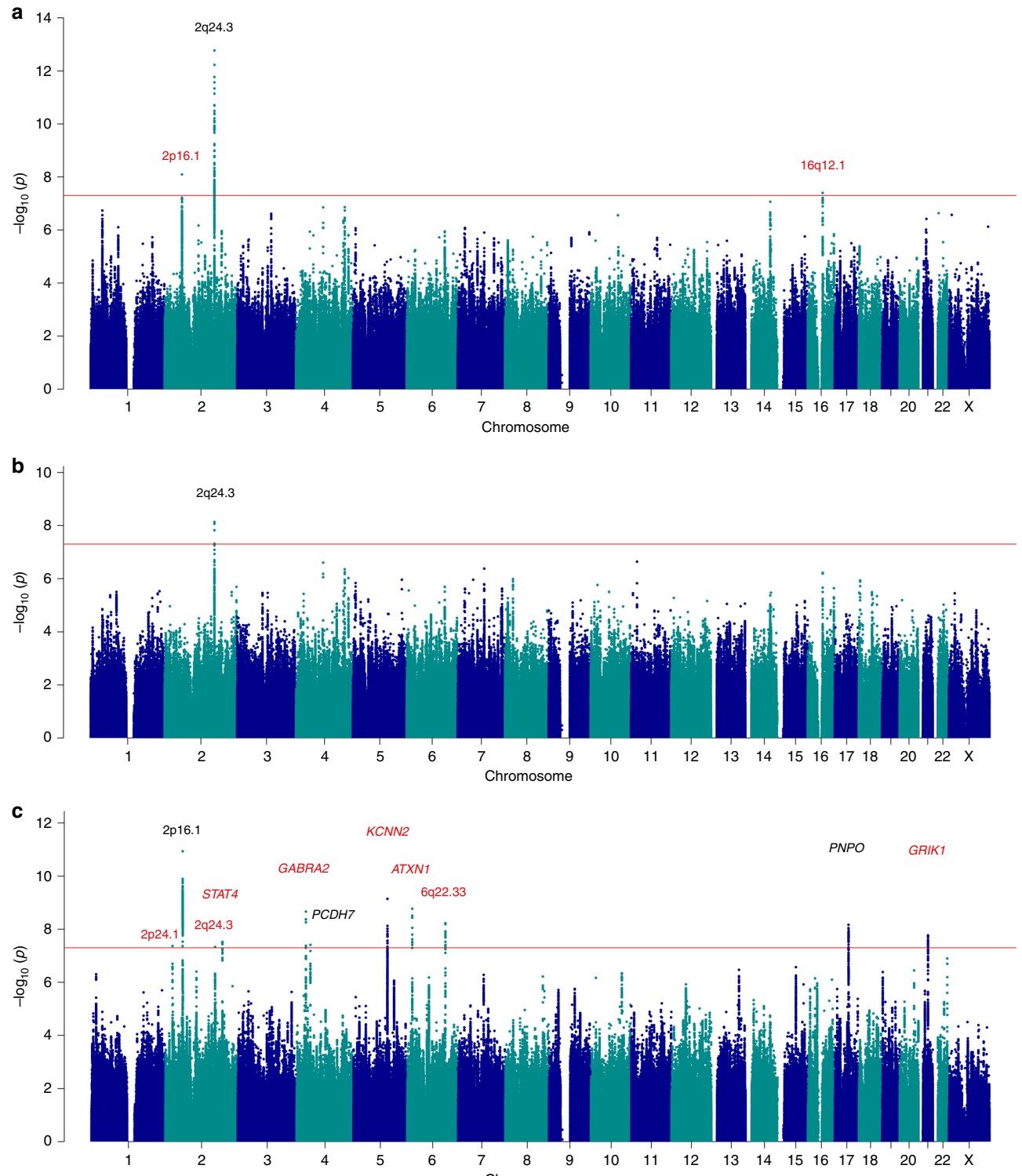

**Fig. 1** Manhattan plots for epilepsy genome-wide association analyses. Genome-wide association analyses of **a** all epilepsy, **b** focal epilepsy, and **c** genetic generalized epilepsy. Negative $\log_{10}$-transformed *P*-values (*Y*-axis) are plotted against chromosomal position (*x*-axis). *P*-values were calculated with METAL using fixed-effects trans-ethnic meta-analyses. The red line represents the genome-wide significance threshold ($p < 5 \times 10^{-8}$). Previously known loci are indicated in black; novel loci in red. The names above each locus represent the prioritized gene in the locus (see Fig. 2) or the name of the locus itself in case of multiple prioritized genes in the locus

transcriptome-wide association study (TWAS) analysis suggests that the signal for genetic generalized epilepsy at 17q21.32, which was also observed in an earlier study[12], seems driven by regulation of expression of *PNPO* (Supplementary Fig. 8). This suggests that

the biology behind pyridoxine (vitamin-B6)-responsive epilepsy[19,20] could be relevant to common genetic generalized epilepsies. Biological prioritization implicates *SCN1A, SCN2A, SCN3A,* and *TTC21B* as the most likely genes underlying the signal at 2q24.3 for

| Phenotype | Locus | Novel/replication | Lead SNP | MAF | Z-score | P-value | Gene | Total score | TWAS | eQTL | Brain exp | Missense | PPI | KO mouse | AED target | Monogenic |
|---|---|---|---|---|---|---|---|---|---|---|---|---|---|---|---|---|
| | | | | | | | | | | | Biological gene criteria | | | | | |
| All epilepsy | 2p16.1 | Replication | rs4671319 (G) | 0.44 | 5.77 | 8.1E−09 | FANCL | 2 | ● | | ● | | | | | |
| | | | | | | | BCL11A | 2 | | | ● | | | ● | | |
| | 2q24.3 | Replication | rs6432877 (G) | 0.26 | 7.37 | 1.7E−13 | SCN3A | 3 | | | ● | | ● | ● | ■ | |
| | | | | | | | SCN2A | 3 | | | ● | | ● | ● | ■ | ■ |
| | | | | | | | TTC21B | 3 | | ● | ● | ● | | | | |
| | | | | | | | SCN1A | 3 | | | ● | | ● | ● | ■ | ■ |
| | 16q12.1 | Novel | rs4638568 (A) | 0.06 | −5.49 | 4.0E−08 | HEATR3 | 2 | | ● | | | ● | | | |
| | | | | | | | BRD7 | 2 | | | ● | | ● | | | |
| Focal epilepsy | 2q24.3 | Replication | rs2212656 (A) | 0.26 | 5.78 | 7.3E−09 | SCN3A | 3 | | | ● | | ● | ● | ■ | |
| | | | | | | | SCN2A | 3 | | | ● | | ● | ● | ■ | ■ |
| | | | | | | | TTC21B | 3 | | ● | ● | ● | | | | |
| | | | | | | | SCN1A | 3 | | | ● | | ● | ● | ■ | ■ |
| Generalized epilepsy | 2p24.1 | Novel | rs4665630 (C) | 0.13 | 5.48 | 4.3E−08 | None | | | | | | | | | |
| | 2p16.1 | Replication | rs1402398 (G) | 0.36 | 6.79 | 1.2E−11 | FANCL | 2 | ● | | ● | | | | | |
| | | | | | | | BCL11A | 2 | | | ● | | ● | ● | | |
| | 2q24.3 | Replication | rs11890028 (G) | 0.27 | −5.46 | 4.7E−08 | SCN3A | 3 | | | ● | | ● | ● | ■ | |
| | | | | | | | SCN2A | 3 | | | ● | | ● | ● | ■ | ■ |
| | | | | | | | TTC21B | 3 | | ● | ● | ● | | | | |
| | | | | | | | SCN1A | 3 | | | ● | | ● | ● | ■ | ■ |
| | 2q32.3 | Novel | rs887696 (C) | 0.34 | 5.54 | 3.0E−08 | STAT4 | 2 | | | ● | | ● | | | |
| | 4p15.1 | Replication | rs1044352 (T) | 0.42 | 5.98 | 2.2E−09 | PCDH7 | 2 | | ● | ● | | | | | |
| | 4p12 | Novel | rs11943905 (T) | 0.27 | 5.50 | 3.9E−08 | GABRA2 | 4 | ● | | ● | | ● | ● | ■ | |
| | 5q22.3 | Novel | rs4596374 (C) | 0.45 | 6.16 | 7.2E−10 | KCNN2 | 2 | | | ● | | | ● | | |
| | 6p22.3 | Novel | rs68082256 (A) | 0.20 | −6.02 | 1.7E−09 | ATXN1 | 2 | | | ● | | ● | | | |
| | 6q22.33 | Novel | rs13200150 (G) | 0.30 | −5.82 | 5.9E−09 | None | | | | | | | | | |
| | 17q21.32 | Replication | rs4794333 (C) | 0.38 | −5.80 | 6.8E−09 | PNPO | 3 | ● | ● | | | ● | | | ■ |
| | 21q22.11 | Novel | rs2833098 (G) | 0.38 | −5.64 | 1.7E−08 | GRIK1 | 2 | | | ● | | ● | | ■ | |
| JME | 16p11.2 | Novel | rs1046276 (T) | 0.34 | 6.67 | 2.5E−11 | STX1B | 4 | ● | ● | ● | | ● | | ■ | |
| CAE | 2p16.1 | Replication | rs12185644 (C) | 0.29 | 6.24 | 4.5E−10 | FANCL | 2 | ● | | ● | | | | | |
| | | | | | | | BCL11A | 2 | | | ● | | ● | ● | | |
| | 2q22.3 | Replication | rs13020210 (G) | 0.20 | −5.58 | 2.4E−08 | ZEB2 | 2 | | | ● | | ● | | | ■ |
| Focal HS | 3q25.31 | Novel | rs1991545 (A) | 0.04 | 6.78 | 1.3E−11 | C3orf33 | 2 | | | | | ● | ● | | |
| | | | | | | | SLC33A1 | 2 | | | | | ● | ● | | |
| | | | | | | | KCNAB1 | 2 | | | ● | | | ● | | |
| | 6q22.31 | Novel | rs1318322 (G) | 0.14 | 5.80 | 6.7E−09 | GJA1 | 2 | | | ● | | ● | | | |

**Fig. 2** Genome-wide significant loci of all analyses and prioritized biological epilepsy genes. Genes were prioritized based on 6 criteria and scored based on the number of criteria met per gene (filled red boxes). The highest scoring gene, or multiple if they have the same score, in each locus is reported as 'prioritized biological epilepsy gene(s)'. Similar to previous studies[17,18], we used a minimum score of 2 to define these genes and we noted 'none' if no gene in the locus reached this score. Filled blue boxes indicate overlap with known targets of anti-epileptic drugs and established monogenic epilepsy genes. The lead SNP is defined as the SNP with the lowest P-value in the locus and the minor allele is displayed in brackets. P-values and Z-scores for All epilepsy, Focal epilepsy and Generalized epilepsy were calculated with fixed-effects trans-ethnic meta-analyses. P-values and Z-scores for JME, CAE, and Focal HS were calculated with BOLT-LMM. MAF minor allele frequency in the Human Reference Consortium reference panel. The direction of the Z-score is signed with respect to the minor allele. TWAS: significant TWAS association (based on data from the CommonMind Consortium), eQTL: significant eQTL within locus (based on data from the ROS/MAP projects), Brain exp: the gene is preferentially expressed in the brain, Missense: epilepsy GWAS missense variant in locus, PPI: gene prioritized by protein-protein interaction, KO mouse: relevant knockout mouse phenotype

all epilepsy, focal epilepsy and genetic generalized epilepsy. Pathogenic variants in the sodium channels *SCN1A*, *SCN2A* and *SCN3A* are associated with various epilepsy syndromes[16] and mutations in *TTC21B* impair forebrain development[21,22]. Our analyses implicate *STX1B* as a potential gene underlying the association of JME at the 16p11.2 locus and the top variant in the locus is an eQTL that strongly correlates with expression of *STX1B* in the dorsolateral prefrontal cortex (Spearman's correlation: Rho $= 0.33$, $p = 3 \times 10^{-14}$)[23]. Interestingly, for one of the prioritized genes in genetic generalized epilepsy, *PCDH7*, an eQTL was recently detected in epileptic hippocampal tissue[24]. Prioritized genes associated with focal epilepsy with hippocampal sclerosis include the gap-junction gene *GJA1*.

In addition we identified eight genes from Fig. 2 (*BCL11A*, *GJA1*, *ATXN1*, *GABRA2*, *KCNAB1*, *SCN3A*, *PCDH7*, *STAT4*) with evidence of co-expression in at least two independent brain expression resources, using a brain gene co-expression analysis with brain-coX[25]. These eight candidates are embedded in several established epilepsy gene co-expression modules (Supplementary Fig. 9; Supplementary Table 9).

**SNP annotation and tissue-specific partitioned heritability**. We functionally annotated all 492 genome-wide significant SNPs from all phenotypes (Fig. 3a–c) and found that most SNPs were either intergenic (29%) or intronic (46%); 78% were in open chromatin regions (as indicated by a minimum chromatin state of 1–7[26,27], and 50% of SNPs showed some evidence for affecting gene transcription (RegulomeDB score ≤6[28]). Four were coding SNPs of which two were missense variants.

To gain further biological insight into our results, we next used a partitioned heritability method[29] to assess whether our genome-wide significant signals contained SNPs associated with enhanced transcription in any of 88 tissues. We found significant enrichment of H3K4me1 markers in all epilepsy (stratified LD-score regression; $p = 4 \times 10^{-5}$) and H3K27ac markers in genetic generalized epilepsy (stratified LD-score regression; $p = 1.3 \times 10^{-6}$), specifically in the dorsolateral prefrontal cortex. Moreover, the distribution of heritability enrichment P-values was strongly skewed towards brain tissues for all epilepsy phenotypes (Fig. 3d, Supplementary Figs. 10–12).

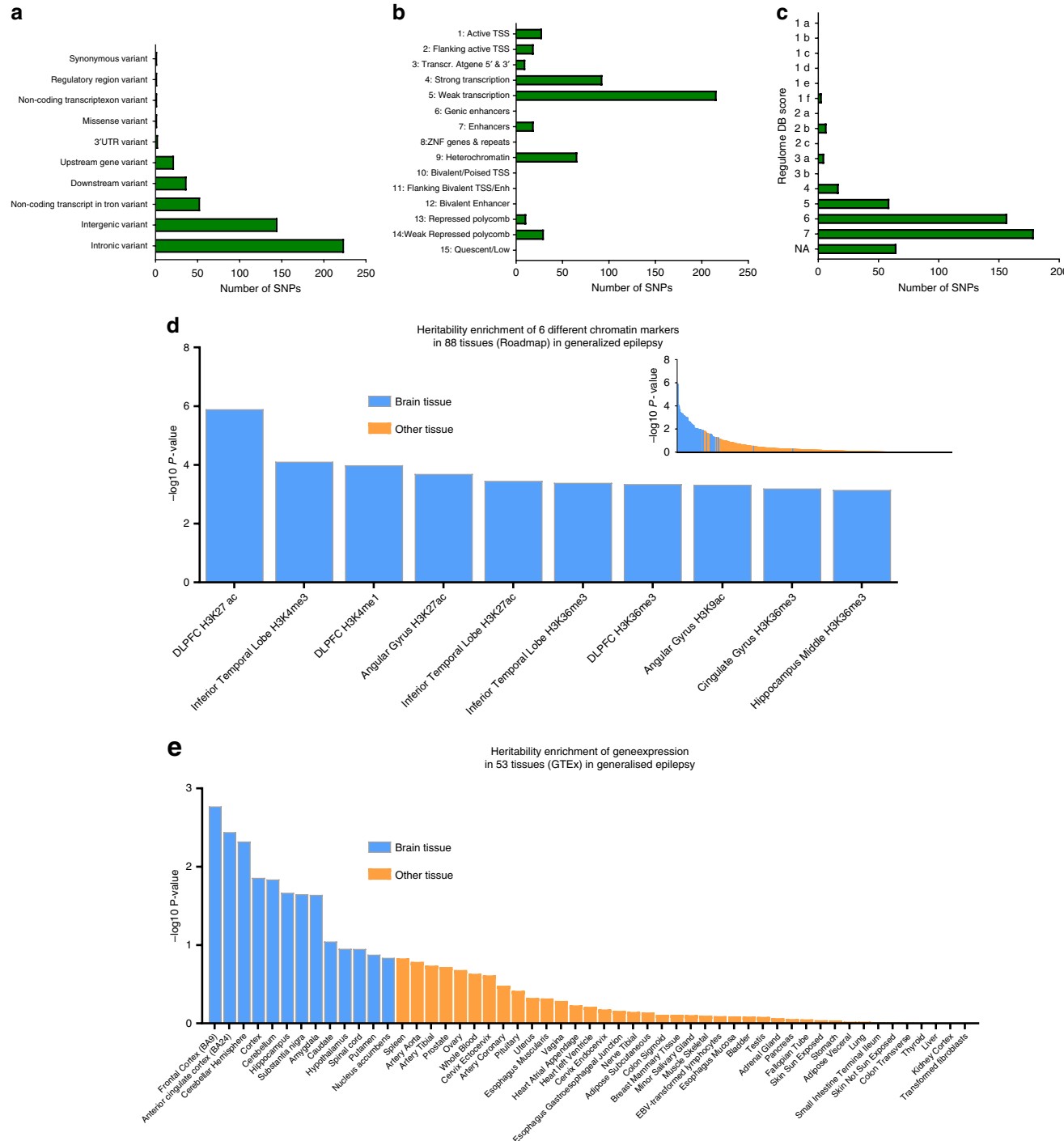

**Fig. 3** Functional annotation and heritability enrichment of epilepsy GWAS results. **a** functional categories of all genome-wide significant SNPs in all phenotypes. **b** Minimum (most active) chromatin state across 127 tissues for all genome-wide significant SNP in all phenotypes; TSS - transcription start site. **c** The RegulomeDB score for all genome-wide significant SNPs in all phenotypes, where 7 represents no evidence for affecting regulation and lower scores represent increasing evidence; NA - the variant does not exist in RegulomeDB. **d** Heritability enrichment for genetic generalized epilepsy with 6 different chromatin markers in 88 tissues, calculated with stratified LD-score regression using data from the Roadmap Epigenomics Project. The main bar chart represent the 10 tissues with the strongest heritability enrichment and the inset shows the full distribution of all chromatin markers in all tissues. **e** Heritability enrichment of genes expressed in 53 tissues, calculated with stratified LD-score regression using data from the gene-tissue expression (GTEx) Consortium

H3K27ac and H3K4me1 are epigenetic markers associated with regulating gene transcription, suggesting that altered transcription in the dorsolateral prefrontal cortex could be one of the underlying mechanisms of epilepsy. This is further supported by a tissue-specific heritability enrichment analysis (using data from the GTEx Consortium), showing strongest enrichment for genetic generalized epilepsy with genes expressed in Brodmann Area 9 (stratified LD-score regression; $p = 1.56 \times 10^{-6}$), which encompasses the dorsolateral prefrontal cortex (Fig. 3e). These findings further corroborate our TWAS results (using data from the unrelated CommonMind Consortium database), which shows significant associations of epilepsy with gene expression of several

genes in the dorsolateral prefrontal cortex (Fig. 2; Supplementary Table 3). Although genetic generalized epilepsy has been regarded as a generalized process, anatomical, electrophysiological, cognitive, and functional imaging studies implicate dysfunction in the frontal lobes[30–34]. Altogether, we have converging evidence from several unrelated methods and databases suggesting epigenetic regulation of gene expression in the dorsolateral prefrontal cortex as a potential pathophysiological mechanism underlying our epilepsy GWAS findings.

Finally, we leveraged the Brainspan database, as implemented in FUMA[35], to assess whether the genes implicated by our GWAS are differentially expressed in the brain at various prenatal and post-natal ages. These analyses were performed for the genes prioritized in any epilepsy phenotype (21 genes), any focal epilepsy subtype (8 genes) or any genetic generalized epilepsy syndrome (15 genes). The results suggest that the expression of genes associated with focal epilepsy is up-regulated in late-infancy and young adulthood, whereas expression of those genes associated with genetic generalized epilepsy is down-regulated in early childhood and differentially expressed prenatally and at adolescence (Supplementary Fig. 13).

**Enrichment analyses.** A previous exome-sequencing study found an association for common epilepsies with ultra-rare variants in known monogenic epilepsy genes[36]. To assess whether common epilepsies are also associated with common variants in monogenic epilepsy genes (see Methods), we pooled the 146 genes that were mapped to our genome-wide significant loci and performed a hypergeometric test. Results illustrated an enrichment of known monogenic epilepsy genes within the genes mapped to our genome-wide significant loci (6 genes overlapped; hypergeometric test: odds ratio [OR] = 8.45, $p = 1.3 \times 10^{-5}$). This enrichment is considerably more significant when limited to the 21 genes with the highest biological priority from Fig 2 (5 genes overlapped; hypergeometric test: OR = 61.4, $p = 9.9 \times 10^{-10}$). We did not find a bias for gene size in our enrichment analyses when using a conservative method to correct for this (see Methods). This suggests that both common and rare variants in monogenic epilepsy genes contribute to common epilepsy susceptibility, corroborating and further extending previous observations[8,37]. Further studies are required to establish whether the signals from common and rare variants are independent of each other.

Using public databases of drug-targets, we found that 13 out of 24 currently licensed anti-epileptic drugs target genes that are implicated in our GWAS. Using the same list of 146 genes as described above, we performed a hypergeometric test which shows a significant enrichment of genes that are known targets of anti-epileptic drugs (8 genes overlapped; hypergeometric test: OR = 19.6, $p = 1.3 \times 10^{-9}$). This enrichment is considerably more significant when limited to the 21 most biologically plausible candidate genes (5 genes overlapped; hypergeometric test: OR = 101.2, $p = 5.7 \times 10^{-11}$). This observation suggests that other drugs that target genes from our GWAS could also have potential for the treatment of epilepsy. The Drug-Gene interaction database (http://dgidb.org) lists 166 drugs that target biologically prioritized genes from our GWAS (see Supplementary Data 2 for a full list), that may be further investigated for their anti-seizure potential.

Next, we used a complementary approach[38] to search for repurposable drugs. By comparing GWAS-imputed and drug-induced transcriptomes, we predicted drugs capable of rectifying epilepsy-associated gene expression changes (see Methods). Our predictions are enriched with licensed antiepileptic compounds (permutation based $p$-value $<1.0 \times 10^{-6}$) and with other licensed compounds that have proven antiepileptic efficacy in animal models (permutation based $p$-value$<1.0 \times 10^{-6}$). We list 30 of our predicted drugs that are licensed for other conditions and have

published evidence of efficacy in animal models of epilepsy (Supplementary Table 4).

**Heritability analyses.** Twin-based and genetic heritability studies have suggested that while epilepsy is strongly heritable[8,39], there is a substantial missing heritability component[40,41]. We used LDAK to estimate $h^2_{SNP}$: the proportion of heritability that can be attributed to SNPs[42–44]. We estimate $h^2_{SNP} = 32.1\%$ (95%CI: 29.6–34.5%) for genetic generalized epilepsy and $h^2_{SNP} = 9.2\%$ (8.4–10.1%) for focal epilepsy (estimates are on the liability scale, assuming a prevalence of 0.002 and 0.003, respectively) which are consistent with previous estimates[8]. These results indicate that SNPs explain a sizeable proportion of the liability of genetic generalized epilepsy syndromes, but less so for focal epilepsy phenotypes (Fig. 4). To delineate the heritability of the different epilepsy phenotypes, we used LDAK to perform genetic correlation analyses between the different forms. We found evidence for strong genetic correlations between the genetic generalized epilepsies, whereas we found no significant correlations between the focal epilepsies (Fig. 4). Interestingly, we found a significant genetic correlation between JME and lesion-negative focal epilepsy (LDAK genetic correlation: $R^2 = 0.46$, $p = 8.77 \times 10^{-4}$), suggesting either pleiotropy and/or misclassification. It is known that focal EEG features can be seen in JME[45].

In view of the increasing data on comorbidities with epilepsy, we next used LD-score regression to analyze the genetic correlation between epilepsy and various other brain diseases and traits from previously published GWAS (Fig. 5; see Supplementary Table 5 for values). Perhaps surprisingly, we did not find significant correlations with febrile seizures. Similarly, we did not find any significant genetic correlations between epilepsy and other neurological or psychiatric diseases. However, we did observe significant correlations for all epilepsy and genetic generalized epilepsy with cognitive ability. We then used the method Multi-Trait Analysis of GWAS (MTAG)[46] to leverage the larger sample size of the genetically correlated GWAS of cognitive ability ($n = 78,308$) in order to boost the effective sample size of our all and genetic generalized epilepsy GWAS to 53,244 and 41,515 respectively. Using this approach, we found a novel genome-wide significant locus at 10q24.32 in all epilepsy (MTAG $p = 2.2 \times 10^{-8}$) and genetic generalized epilepsy (MTAG $p = 4.0 \times 10^{-8}$) which encompasses the $K_v$-channel-interacting protein 2 (KCNIP2) gene (Supplementary Fig. 14), loss of which is associated with seizure susceptibility in mice[47].

**Discussion**

The increased sample size in this second ILAE Consortium GWAS of common epilepsies has resulted in the detection of 16 risk loci for epilepsy and illustrates how common variants play an important role in the susceptibility of these conditions. But compared to other common neurological diseases our sample size is modest. For example the latest GWAS in schizophrenia considered 36,989 schizophrenia cases and 113,075 controls, resulting in the identification of 108 risk loci[48]. Larger efforts would deliver further insight to the genetic architecture of the common epilepsies.

The majority of the loci are associated with genetic generalized epilepsy. This observation is a welcome partial explanation for the high heritability of genetic generalized epilepsy, in light of the relative lack of rare variant variants discovered to date. We also show that there is substantial genetic correlation between the generalized syndromes. We speculate that the subtypes share a large part of the genetic susceptibility for generalized epilepsies, with specific modifying factors determining the specific syndrome.

**Fig. 4** Heritability estimates and genetic correlations between epilepsy syndromes, calculated using LDAK. Subjects with a diagnosis of both CAE and JAE were excluded from both phenotypes. The genetic correlation coefficient was calculated with LDAK and is denoted with a color scale ranging from 0% (white) to 100% (red). #P < 0.05; *P < 0.0024 (Bonferroni threshold); $h_L^2$2 SNP-based heritability on liability scale (95% CI); †heritability estimate exceeded 100%, possibly due to small sample size and large SD; CAE - childhood absence epilepsy, JAE - juvenile absence epilepsy, JME - juvenile myoclonic epilepsy, GTCS alone - generalized tonic-clonic seizures alone, focal HS - focal epilepsy with hippocampal sclerosis

**Fig. 5** Genetic correlations of epilepsy with other phenotypes. The genetic correlation coefficient, calculated using LD-score regression, is denoted with a color scale ranging from -100% (blue) to 100% (red). #: P < 0.05 *P < 0.001 (Bonferroni threshold; 0.05/48)

Some syndrome-specific associations were detected, such as the relatively strong signal for *STX1B* in JME, and the association of *GJA1* with focal epilepsy-hippocampal sclerosis. Interestingly, although the association signal for *STX1B* was only significant in the JME analysis, rare pathogenic variants in *STX1B* have been recently found in a spectrum of epilepsies, including genetic epilepsy with febrile seizures plus (GEFS+), genetic generalized epilepsies (including JME), epileptic encephalopathies and even some focal epilepsies[49,50] (Wolking et al., Manuscript submitted (2018). Further, mutations in the gap-junction gene *GJA1* are associated with impaired development of the hippocampus[51] and different expression has been reported in epileptic hippocampal and cortical tissue[52,53]. These findings represent a tantalizing glance of the different biological mechanisms underlying epilepsy syndromes that may guide us to the introduction of genetics for improved diagnosis, prognosis and treatment for these common epilepsies. However, the relatively low sample size of our subtype analysis warrants a conservative interpretation and follow-up with a larger cohort.

At least three association signals are shared between focal epilepsy and genetic generalized epilepsy. The clearest overlapping signal remains the 2q24.3 locus, as we reported previously[15]. However, this association signal is complex and we demonstrate that the locus consists of at least two independent signals (Supplementary Fig. 3). Our Hi-C chromatin analysis suggests the complexity includes levels of functional association to *SCN2A* and *SCN3A*, that are located more distally to the *SCN1A* locus. Mutations in *SCN2A* and more recently *SCN3A* are established monogenic causes of epileptic encephalopathy that, like *SCN1A*, cause dysfunction of the encoded ion-channels, which is believed to disturb the fine balance between neuronal excitation and inhibition. This may involve independent variation that either affects regulation of *SCN1A*, *SCN2A*, or *SCN3A* independently. However, the complex association may also reflect multiple rare risk variations, and large resequencing studies will shed further light on this issue.

The number of association signals we detected and increased power relative to our previous meta analysis[15] allowed us to explore the biological mechanisms behind the observed genetic associations. We show that the signals converge on the dorsolateral prefrontal cortex as the tissue in which most functional effect is observed; this is broadly consistent with the importance of the frontal lobes in generalized epilepsies. Indeed, our analyses of the epigenetic markers H3K27ac and H3K4me1, TWAS, and tissue-specific heritability enrichment analysis all point towards epigenetic regulation of gene expression in the dorsolateral prefrontal cortex as a potential pathophysiological mechanism underlying our epilepsy GWAS findings.

Altogether, we found 16 loci that are associated with the common epilepsies. Our heritability analyses show that collectively, common genetic variants explain a third of the liability for genetic generalized epilepsy. Our analyses suggest that the associated variants are involved in regulation of gene expression in the brain. The 21 biological epilepsy candidate genes implicated by our study have diverse biological functions, and we show that these are enriched for known epilepsy genes and targets of current antiepileptic drugs. Our analyses provide evidence for pleiotropic genetic effects that raise risk for the common epilepsies collectively, as well as variants that raise risk for specific epilepsy syndromes. Determining the shared and unique genetic basis of epilepsy syndromes should be of benefit for further stratification and eventually lead to possible applications for improved diagnosis, prognosis, and treatment. Future studies including pharmacoresponse data, imaging, and other clinical measurements have the potential to further increase the benefit of these studies for people with epilepsy. In combination, these findings further our understanding of the complex genetic architecture of the epilepsies and could provide leads for new treatments and drug repurposing.

## Methods

**Ethics statement**. We have complied with all relevant ethical regulations. All study participants provided written, informed consent for use of their data in genetic studies of epilepsy. For minors, written informed consent was obtained from their parents or legal guardian. Local institutional review boards approved study protocols at each contributing site.

**Cohorts and phenotype definition**. A list of the sites included in this study is described in Supplementary Table 6. We classified seizures and epilepsy syndromes according to the classification and terminology outlined by the ILAE[15,54]. For all cases, epilepsy specialists assessed each phenotype at the contributing site. Individuals with epilepsy were initially assigned to one of three phenotypic categories: genetic generalized epilepsy, focal epilepsy, or unclassified epilepsy. Based on EEG, MRI and clinical histories we further classified our cohort into the epilepsy subtypes listed in Supplementary Table 1. We used a combination of population-based datasets as controls. Some control cohorts were screened by questionnaire for neurological disorders. 53.4% of cases were female compared to 51.6% of controls.

**Study design**. We conducted a case-control study in subjects of Caucasian, Asian (Han Chinese) and African-American ethnicities. Our primary analyses were structured to test common genetic variants for association with epilepsy according to broad epilepsy phenotypes. We pooled cases from cohorts of the same ethnic group to perform linear mixed model analysis, followed by subsequent meta-analyses of regression coefficients across the three ethnic groups. Our secondary analyses tested for associations with specific syndromes of genetic generalized epilepsy (childhood absence epilepsy, juvenile absence epilepsy, juvenile myoclonic epilepsy, and generalized tonic-clonic seizures alone) and phenotypes of focal epilepsy (lesion negative, focal epilepsy with hippocampal sclerosis, and focal epilepsy with other lesions). The secondary analyses were limited to Caucasian subjects due to sample size. We prioritized the results of the GWAS by incorporating eQTL information, transcriptome-wide analysis, and biological annotation. Finally, we estimated the genetic correlation of epilepsy phenotypes using Linkage-Disequilibrium Adjusted Kinships (LDAK).

**Genotyping**. The EpiPGX samples were genotyped at deCODE Genetics on Illumina OmniExpress-12 v1.1 and OmniExpress-24 v1.1 single nucleotide polymorphism (SNP) arrays. The EPGP samples were genotyped on Illumina HumanCore beadchips at Duke University, North Carolina. The remainder of the samples were genotyped on various SNP arrays, as previously published[15].

**Genotyping quality control and imputation**. Quality control of genotyping was performed separately for each cohort using PLINK 1.9[55]. Each genotype cohort was temporarily merged with a genetically similar reference population from the 1000 Genomes Project (CEU, CHB, or YRI). A test for Hardy–Weinberg equilibrium (HWE) was performed and SNPs significant at $p < 1 \times 10^{-10}$ were removed. All samples and all SNPs with missing genotype rate >0.05 and all SNPs with minor allele frequency (MAF) <0.01 were removed. Next, we pruned SNPs using the PLINK --indep-pairwise command (settings: window size 100 kb, step size 25, $R^2 > 0.1$). Using this subset of SNPs, we removed samples with outlying heterozygosity values (>5 SD from the median of the whole cohort). Identity by descent (IBD) was calculated in PLINK to remove sample duplicates (>0.9 IBD) and to identify cryptic relatedness. We removed one from each sample pair with IBD>0.1875, with the exception of the EPGP familial epilepsy cohort. Subjects were removed if sex determined from X-chromosome genotype did not match reported gender. Array-specific maps were used to update all SNPs positions and chromosome numbers to the Genome Reference Consortium Human Build 37 (GRCh37), and remove all A/T and C/G SNPs to avoid strand issues. We applied pre-imputation checks according to scripts available on the website of Will Rayner of the Wellcome Trust Centre for Human Genetics (www.well.ox.ac.uk/~wrayner/tools/) to remove SNPs with allele frequencies deviating >20% from the frequency in the Haplotype Reference Consortium. Samples were submitted to the Sanger Imputation Service (https://imputation.sanger.ac.uk/)[56]. We selected the Human Reference Consortium (release 1.1; $n = 32470$) dataset as reference panel for Caucasian and Asian datasets and the African Genome Resources ($n = 4956$) for the African-American datasets. Post-imputation quality control filters were applied to remove SNPs within each imputed cohort with an imputation info score <0.9 or HWE $p<1e-6$. Imputed genotype dosages with a minimum certainty of 0.9 per subject were converted to hard-coded PLINK format after which all samples were pooled into a single cohort. We performed a principal components analysis using GCTA. From the PCA results we stratified our subjects into three broad ethnic groups (Caucasian, Asian and African) while removing extreme outliers. After stratifying by ethnicity, we removed SNPs with HWE $p < 1e-6$, call rate <0.95 or MAF<0.01. In total 816 subjects out of 45705 subjects were filtered out by quality control procedures, leaving 44889 subjects for analyses.

**Study power**. We estimated using PGA[57] that the study had 80% power to detect a genetic predictor of relative risk for all epilepsy (approximated to odds ratio) ≥1.45 with MAF = 1% and an alpha level of $5 \times 10^{-8}$. We estimated that our meta-analyses had 80% power to detect genome-wide significant SNPs of MAF = 1% with relative risks ≥1.5 and ≥1.8, for focal and generalized epilepsy respectively (see Supplementary Figure 15). Our analysis of generalized epilepsy sub-phenotypes had 80% power to detect genome-wide significant SNPs of MAF = 1% with relative risks ≥2.6, ≥3.3, and ≥2.4 for CAE, JAE, and JME respectively. Our analysis of focal epilepsy sub-phenotypes had 80% power to detect genome-wide significant SNPs of MAF = 1% with relative risks ≥1.9, ≥2.8, and ≥1.9 for focal epilepsy lesion negative, focal epilepsy with hippocampal sclerosis and focal epilepsy with lesion other than hippocampal sclerosis, respectively.

**Statistical analyses**. Association analyses were conducted within the three ethnic subgroups using a linear mixed model in BOLT-LMM[58]. A subset of SNPs, used to correct for (cryptic) relatedness and population stratification by BOLT-LMM, were derived by applying SNP imputation info score >0.99, MAF >0.01, call rate >0.99 before pruning the remaining variants using LDAK with a window size of 1 Mb and $R^2 > 0.2$[43]. All analyses included gender as a covariate and the threshold for statistical significance was set at $5 \times 10^{-8}$. We compared $\chi^2$ values of the BOLT-LMM output between all pairs of SNPs in high LD ($R^2 > 0.4$) and removed pairs of SNPs with extreme $\chi^2$ differences using a formula that scales exponentially with magnitude of $\chi^2$ and LD: $\chi^2$ difference cutoff $= \frac{3 * \sqrt{\frac{SNP1-\chi2+SNP2-\chi2}{2}}}{(R^2)^2}$; where $SNP1$-$\chi^2$ and $SNP2$-$\chi^2$ are the $\chi^2$-statistic of the two SNPs in each pair and $R^2$ is their squared correlation (LD). We tested the homogeneity of all SNPs by splitting the pooled cohort into 13 distinct clusters of ethnically matched cases and controls and removed SNPs exhibiting significant heterogeneity of effect ($P_{het} < 1 \times 10^{-8}$). Fixed effects, trans-ethnic meta-analyses were conducted using the software package METAL[59]. Manhattan plots for all analyses were created using qqman. Considering that our study had unequal case-control ratios, we calculated the effective sample size per ethnicity using the formula recommended by METAL: $N_{eff} = 4/(1/N_{cases} + 1/N_{ctrls})$. Since >95% of all cases were Caucasian, we included all SNPs that were present in at least the Caucasian dataset (~5 million).

Conditional association analysis was performed with PLINK on loci containing significant SNPs to establish whether other genetic variants in the region (500 Kb upstream and downstream) were independently associated with the same phenotype. The conditional threshold for significance was set at $2 \times 10^{-5}$, based on approximately 2500 imputed variants per 1MB region.

**Assessment of inflation of the test statistic**. Potential inflation of the test statistic was assessed per ethnicity and phenotype by calculating the genomic inflation factor ($\lambda$; the ratio of the median of the empirically observed distribution of the test statistic to the expected median) and the mean $\chi^2$. Since $\lambda$ is known to scale with sample size, we also calculated the $\lambda_{1000}$, i.e $\lambda$ corrected for an equivalent sample size of 1000 cases and 1000 controls[60]. We observed some inflation of the test statistic ($\lambda > 1$) across the different phenotypes (Supplementary Table 7), suggesting either polygenicity or confounding due to population stratification or cryptic relatedness. Therefore, we applied LD score regression[61], estimating LD scores using matched populations from the 1000 GP (EUR for Caucasians ($n = 669$), AFR for African-Americans and EAS for Asians). These LDSC results suggested that inflation of the test statistic was primarily due to polygenicity for most analyses (Supplementary Table 7). Only the Caucasian focal and all epilepsy analyses had LDSC intercepts >1.1, suggesting confounding or an incomplete match of the LD-score reference panel. Our focal and all epilepsy analyses included cohorts from various Caucasian ethnicities, including Finnish and Italian focal epilepsy cohorts, and it has been shown that LD differs considerably between Finnish and Italian populations[61]. Therefore, we consider an incomplete match of the LD-score reference panel the most likely cause of the observed inflation, since we used a mixed-model analysis that corrects for population stratification and cryptic relatedness[58].

**Gene mapping and biological prioritization**. Genome-wide significant loci of all phenotypes were mapped to genes in and around these loci using FUMA[35]. Genome-wide significant loci were defined as the region encompassing all SNPs with $P < 10^{-4}$ that were in LD ($R^2 > 0.2$) with the lead SNP (i.e. the SNP with the lowest P-value in the locus with $P < 5 \times 10^{-8}$). Positional mapping was performed to map genes that were located within 250 kb of these loci. Additionally, we mapped genes that were farther than 250 kb away from the locus using chromatin interaction data to identify genes that show a significant 3D interaction ($P_{FDR} < 10^{-6}$) between their promoter and the locus, based on Hi-C data from dorsolateral prefrontal cortex, hippocampus, and neural progenitor cells[62]. This resulted in a total of 146 mapped genes across all phenotypes, of which some genes (e.g. SCN1A) were associated with multiple epilepsy phenotypes.

We next devised various prioritization criteria to prioritize the most likely biological candidate genes out of the 146 mapped genes, similar to previous studies[17,18,63], based on the following 6 criteria:

1. A significant correlation between the epilepsy phenotype and expression of the gene, as assessed with a transcriptome-wide association study (TWAS). Default settings of the FUSION software package[64] were used to impute gene-expression based on our GWAS summary statistics and RNA-sequencing data from dorsolateral prefrontal cortex tissue ($n = 452$, CommonMind Consortium[65]), after which the association between the epilepsy phenotype with gene-expression was calculated. It was possible to test the TWAS expression association for 53 out of our 146 mapped genes, since only the expression of these 53 genes was significantly heritable (heritability $p$-value <0.01, as suggested by Gusev et al.[64]). We set a Bonferroni corrected $p$-value threshold of 0.05/53 = 0.00094 to define significant TWAS associations.

2. Genes for which a SNP in the genome-wide significant locus (as defined above) is a significant cis-eQTL (Bonferroni corrected $P < 8 \times 10^{-10}$)[23] based on data from the ROS and MAP studies, which includes RNA-sequencing data from 494 dorsolateral prefrontal cortex tissues[23].

3. The gene is preferentially expressed in the brain. This was assessed by using gene-expression data from all 53 tissues of the Gene-Tissue expression (GTEx) Consortium[66]. Genes were considered to be preferentially expressed in the brain when the average expression in all brain tissues was higher than the average expression in non-brain tissues.

4. Genes for which a SNP in the genome-wide significant locus (as defined above) is a missense variant, as annotated by ENSEMBL[67].

5. Genes prioritized by protein-protein interaction network, as calculated using the default settings of DAPPLE[68], which utilizes protein–protein interaction data from the InWeb database[69]. The 146 genes implicated by our GWAS were input after which DAPPLE assessed direct and indirect physical interactions to create a protein-interaction network. Next, DAPPLE assigned a significance score to each gene according to several connectivity parameters; genes with a corrected $P < 0.05$ were considered to be prioritized by DAPPLE.

6. Genes for which a nervous system or behavior/neurological phenotype was observed in knockout mice. Phenotype data of knockout mice was downloaded from the Mouse Genome Informatics database (http://www.informatics.jax.org/) on 17 January 2018 and nervous system (phenotype ID: MP:0003631) and behavior/neurological phenotype (MP:0005386) data were extracted.

All 146 genes were scored based on the number of criteria met (range 0–6 with an equal weight of 1 per criterion), see Supplementary Data 1 for a full list. We considered the gene(s) with the highest score in each locus as the most likely biological epilepsy candidate gene. Multiple genes in a locus were selected if they had an equally high score whilst no genes were selected in a locus if all genes within it had a score <2, similar to previous studies[17,18].

**Gene co-expression analysis for epilepsy with brain-coX**. In silico gene prioritization was performed using brain-coX[25]. brain-coX uses a compendium of seven large-scale normal brain gene expression data resources to identify co-expressed genes with a set of given genes (known, or putative, disease causing genes) likely to encapsulate gene expression networks involved in disease. This approach can identify, and thus leverage networks that are not currently known and not present in available resources such as PPI networks and is a complementary approach to these. We used a set 102 monogenic epilepsy genes (Supplementary Table 8) as the set of known disease genes. An FDR of 0.2 was used to identify genes that significantly co-express with the known set of genes. Prioritization in at least three datasets at an FDR of 0.2 led to a specificity of 0.9[25].

In the first analysis we used a set of 16 candidate epilepsy genes identified by the GWAS analysis and prioritized using additional methods (Fig. 2). These excluded any genes already included in the set of known epilepsy genes (Supplementary Table 8). Supplementary Fig. 9 shows the gene co-expression pattern using the weighted average gene co-expression across all seven datasets for candidate genes from the GWAS that show significant gene co-expression with any of the 102 known epilepsy genes.

In the second analysis we used the set of all the 146 candidate genes identified in the GWAS analysis (Supplementary Data 1). Only 140 of these were identified as having available expression data in the gene expression resources. Many genes showed some evidence of gene co-expression but few showed co-expression in more than 2 datasets (18 out of 140). BCL11A (6) and GJA (6) remain the most robust candidate genes co-expressed with known epilepsy genes. The complete results are shown in Supplementary Table 9.

**Functional annotations**. We annotated all genome-wide significant SNPs ($p < 5 \times 10^{-8}$) from all phenotypes using the Variant Effect Predictor of ENSEMBL[67] and the RegulomeDB database[28]. We annotated chromatin states using epigenetic data from the NIH Roadmap Epigenomics Mapping Consortium[70] and ENCODE[71]. We used FUMA[35] to annotate the minimum chromatin state (i.e. the most active state) across 127 tissues and cell types for each SNP, similar to a previous study[27].

**Heritability enrichment of epigenetic markers and gene-expression**. We used stratified LD-score regression[72] to assess tissue-specific heritability enrichment of epigenetic markers in 88 tissues, using standard procedures[29]. We used the same

settings and pre-calculated weights that accompanied the paper by Finucane et al. to calculate the heritability enrichment of all epilepsy, focal epilepsy and generalized epilepsy, based on epigenetic data of 6 chromatin markers in 88 tissues from the Roadmap Consortium and gene-expression data in 53 tissues from the GTEx Consortium.

**Enrichment analyses.** Hypergeometric tests were performed with R (version 3.4.0) to assess whether the genes mapped to genome-wide significant loci and the subset of prioritized biological epilepsy genes (see above) were enriched for: (i) known monogenic epilepsy genes ($n = 102$) and (ii) known anti-epileptic drug target genes ($n = 64$), relative to the rest of the protein-coding genes in the genome ($n = 19180$). We supplemented the list of 43 known dominant epilepsy genes[36] with an additional 59 monogenic epilepsy genes from the GeneDX comprehensive epilepsy panel (www.genedx.com). We compiled the list of drug target genes from[73], supplemented with additional FDA & EMA licensed AEDs. The full list of gene targets considered in each analysis are listed in Supplementary Tables 8 and 10.

**Enrichment analyses corrected for gene size.** Brain expressed genes are known to be larger in size than non-brain expressed genes. To assess whether gene size could be a cause of bias for our enrichment analyses, we first assessed whether the size of the genes mapped in our analyses was different than non-mapped genes in the genome. We found that the size of the 146 genes mapped to genome-wide significant loci was 65.6 kb, whereas the average gene size of all other protein-coding genes is on average 62.2 kb, suggesting there is no strong bias towards preferentially mapping loci to small or large genes.

We also observed that the 102 established monogenic epilepsy genes are on average 2.44 times longer than non-epilepsy genes (152.0 kb vs 62.2 kb). As a conservative approach to correct for this size difference, we have used the Wallenius' noncentral hypergeometric distribution, as implemented in the R-package 'BiasedUrn'. Using this distribution, we repeated our hypergeometric analyses under the conservative assumption of a 2.42 times increased likelihood of mapping epilepsy genes as opposed to non-epilepsy genes. Using this distribution, the 146 genes that were mapped to genome-wide significant loci were significantly enriched for monogenic epilepsy genes (Wallenius' noncentral hypergeometric test $p = 8.3 \times 10^{-3}$). When limiting our results to the 21 biological prioritized genes, the enrichment of monogenic epilepsy genes became more significant (Wallenius' noncentral hypergeometric distribution $p = 5.3 \times 10^{-4}$).

Similarly, we observed that the targets of AEDs are on average 2.43 times longer than non-AED target genes (151.8 kb vs 62.4 kb). When correcting for this gene-size difference under the assumption of a 2.43 times increased likelihood of mapping our genome-wide significant loci to AED target genes, we find that the 146 mapped genes were significantly enriched for AED target genes (Wallenius' noncentral hypergeometric test $p = 1.7 \times 10^{-5}$). When limiting our results to the 21 biological prioritized genes, the enrichment of AED target genes became more significant (Wallenius' noncentral hypergeometric test $p = 1.0 \times 10^{-8}$).

**Connectivity mapping.** Connectivity mapping was performed using our GWAS results in order to identify drugs which can potentially be repurposed for the treatment of epilepsy, enabling significant savings in the time and cost of anti-epileptic drug development. Recently, So et al. identified candidate drugs that could be repurposed for the treatment of schizophrenia by using GWAS results to impute the gene-expression changes associated with the disease and, then, identifying drugs that change gene-expression in the opposite direction in cell lines[38]. Interestingly, the set of candidate drugs they identified was significantly enriched with antipsychotics. We adopted a similar strategy.

Gene-expression changes associated with epilepsy were imputed from the all epilepsy GWAS summary statistics using the FUSION software package[64] and dorsolateral prefrontal cortex tissue RNA-sequencing data ($n = 452$, CommonMind Consortium[65]). We calculated z-scores for the association between epilepsy and changes in expression of all 5261 significantly heritable genes, using default settings of the FUSION software package as described above[64]. The top 10% of the gene-expression changes most strongly associated with epilepsy were used to construct the disease signature. Then, we identified drugs that change gene-expression in the opposite direction in cell lines, using the Combination Connectivity Mapping bioconductor package and the Library of Integrated Network-Based Cellular Signatures (LINCS) data[74]. This package utilizes cosine distance as the (dis)similarity metric[75,76]. A higher (more negative) cosine distance value indicates that the drug induces gene-expression changes more strongly opposed to those associated with the disease. A lower (more positive) cosine distance value indicates that the drug induces gene-expression changes more similar to those associated with the disease. In the LINCS dataset, some drugs have been profiled in more than one cell line, concentration, and time-point. For such drugs, the highest absolute cosine distance, whether positive or negative, was selected, as this value is less likely to occur by chance. The output of this analysis comprised 24,051 drugs or 'perturbagens', each with a unique cosine distance value.

To demarcate the set of drugs predicted to significantly reverse epilepsy-associated gene-expression changes, the threshold of statistical significance for cosine distance values was determined. For this, we performed 100 permutations of the disease gene-expression z-scores and compared them to drug gene-expression

signatures. We combined the distribution of cosine distance values across all permutations, such that the null distribution was derived from 2,405,100 cosine distance values under $H_0$. The cosine distance value corresponding to α of 0.05 was −0.386. Of the drugs with a cosine distance less than −0.386, thirty were experimentally-validated drug repurposing candidates from the Prescribable Drugs with Efficacy in Experimental Epilepsies (PDE3) database—a recently published systematic and comprehensive compilation of licenced drugs with evidence of antiepileptic efficacy in animal models[77]. We determined whether this is more than expected by chance, by creating 1,000,000 random drug-sets of the same size as drugs with a significant cosine distance. Next, we counted the number of subsets containing an equal or higher number of experimentally-validated drug repurposing candidates, as those found within drugs with a significant cosine distance. This permutation-based $p$-value was $1.0 \times 10^{-6}$.

Supplementary Table 4 lists the 30 candidate re-purposable drugs that are predicted to significantly reverse epilepsy-associated gene-expression changes, have published evidence of antiepileptic efficacy in animal models, and are already licensed for the treatment of other human diseases. Of this list, 22 drugs have corroborated evidence of antiepileptic efficacy from multiple published studies or multiple animal models. For each drug, we list the studies demonstrating antiepileptic efficacy in animal models, the animal models used, and the licensed indication(s).

**Validation of connectivity mapping results.** Validation of the connectivity mapping results was performed using two non-overlapping sets of drugs with known antiepileptic efficacy: (1) a set of 'clinically-effective' drugs that have antiepileptic efficacy in people, and (2) a set of 'experimentally-validated' drugs that have antiepileptic efficacy in animal models. For the clinically-effective drug-set, we used the names of all recognized antiepileptic drugs, as listed in category N03A of the World Health Organization (WHO) Anatomical Therapeutic Chemical (ATC) Classification System, and of benzodiazepines and their derivatives (ATC codes N05BA and N05CD), and of barbiturates (ATC code N05CA), as these drugs are known to have antiepileptic efficacy in people. For the experimentally-validated drug-set, we extracted drug names from the PDE3 database[77].

We determined whether, in our results, clinically effective drugs are ranked higher than expected by chance. The median rank of all drugs was 12,026. The median rank of clinically effective drugs was 3725. Hence, the median rank of clinically-effective drugs was 8301 positions higher than that of all drugs. A permutation-based $p$-value was determined by calculating the median ranks of 1,000,000 random drug-sets, each equal to the number of clinically effective drugs in the LINCS database. This permutation-based $p$-value was $<1.0 \times 10^{-6}$. Similarly, we determined whether, in our results, experimentally-validated drugs are ranked higher than expected by chance. The median rank of experimentally-validated drugs was 6372. Hence, the median rank of experimentally-validated drugs was 5654 positions higher than that of all drugs. A permutation-based $p$-value was determined by calculating the median ranks of 1,000,000 random drug-sets, each equal in size to the number of experimentally-validated drug repurposing candidates in the LINCS database. This permutation-based $p$-value was $<1.0 \times 10^{-6}$.

**Heritability analysis.** Linkage-Disequilibrium Adjusted Kinships (LDAK[42,43]) was used to calculate SNP-based heritability of all epilepsy phenotypes. Since these analyses require homogeneous cohorts, only Caucasian subjects (which represent >95% of epilepsy cases) were used for these analyses. SNP based heritabilities ($h_o^2$) were converted to liability scale heritability estimates ($h_L^2$) using the formula:[8] $h_L^2 = h_o^2 * K^2(1-K)^2/p(1-p) * Z^2$, where $K$ is the disease prevalence, $p$ is the proportion of cases in the sample, and $Z$ is the standard normal density at the liability threshold. We estimated disease prevalence based on a combination of previous studies[8,78,79] (Supplementary Table 11). Although prevalence estimates vary between studies, the $h_L^2$ estimate has been shown to be fairly robust to such differences[8]. Similarly, we have modeled $h_L^2$ using half and double of our prevalence estimates which lead to $h_L^2$ estimates that varied between 0.4 and 11% (Supplementary Table 11). In addition, we compared the heritability estimates from LDAK with the alternative methods BOLT-REML[80] and LDSC[58] (Supplementary Table 12). Next, LDAK was used to calculate the genetic correlation between the 7 epilepsy subtypes. Subjects with a diagnosis of both CAE and JAE were excluded from heritability and genetic correlation analyses.

We computed the genetic correlation between all, focal and genetic generalized epilepsy with other brain diseases and traits using LDSC, as implemented in LD hub[81]. LD hub is a centralized database that contains publicly available GWAS summary statistics from various diseases and traits. We selected published GWAS of psychiatric, neurological, auto-immune diseases with known brain involvement and cognitive/behavioral traits from LD hub. We contacted the authors of published GWAS to provide us with summary statistics when no summary statistics were available on LDhub or when a more recent GWAS of a disease/trait was published that was not included in LDhub. The Caucasian subset of our data was used for all analyses and only other GWAS with primarily Caucasian subjects were included in our analyses. We used a conservative Bonferroni correction to assess significance of genetic correlations ($p = 0.05/48 = 0.001$).

Multi-trait analysis of GWAS (MTAG[46]) was used with default settings to increase the effective sample size from our Caucasian all and generalized epilepsy GWAS by pairing it with the significantly correlated GWAS on cognitive ability (as

assessed above) with a larger sample size ($n$=78,307). MTAG utilizes the fact that estimations of effect size and standard error of a primary GWAS, in this case epilepsy, can be improved by matching them to a genetically correlated secondary GWAS, in this case cognitive ability.

## Data availability

The GWAS summary statistics data that support the findings of this study are available at http://www.epigad.org/gwas_ilae2018_16loci.html.

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

## Acknowledgements

We are grateful to the patients and volunteers who participated in this research. We thank the following clinicians and research scientists for their contribution through sample collection (cases and controls), data analysis, and project support: Geka Ackerhans, Muna Alwaidh, R E Appleton, Willem Frans Arts, Guiliano Avanzini, Paul Boon, Sarah Borror, Kees Braun, Oebele Brouwer, Hans Carpay, Karen Carter, Peter Cleland, Oliver C Cockerell, Paul Cooper, Celia Cramp, Emily de los Reyes, Chris French, Catharine Freyer, William Gallentine, Michel Georges, Peter Goulding, Micheline Gravel, Rhian Gwilliam, Lori Hamiwka, Steven J Howell, Adrian Hughes, Aatif Husain, Monica Islam, Floor Jansen, Mary Karn, Mark Kellett, Ditte B Kjelgaard, Karl Martin Klein, Donna Kring, Annie WC Kung, Mark Lawden, Jo Ellen Lee, Benjamin Legros, Leanne Lehwald, Edouard Louis, Colin HT Lui, Zelko Matkovic, Jennifer McKinney, Brendan McLean, Mohamad Mikati, Bethanie Morgan-Followell, Wim Van Paesschen, Anup Patel, Manuela Pendziwiat, Marcus Reuber, Richard Roberts, Guy Rouleau, Cathy Schumer, B Sharack, Kevin Shianna, NC Sin, Saurabh Sinha, Laurel Slaughter, Sally Steward, Deborah Terry, Chang-Yong Tsao, TH Tsoi, Patrick Tugendhaft, Jaime-Dawn Twanow, Jorge Vidaurre, Sarah Weckhuysen, Pedro Weisleder, Kathleen White, Virginia Wong, Raju Yerra, Jacqueline Yinger and all contributing clinicians from the Department of Clinical and Experimental Epilepsy at the National Hospital for Neurology and Neurosurgery and UCL Institute of Neurology. Data generated as part of the EPIGEN Consortium was included in this study. We would like to thank Dr. Weihua Meng (University of Dundee), Dr. Mark Adams (University of Edinburgh) and Dr. Ynte Ruigrok (UMC Utrecht), Dr. Bjarke Feenstra (Statens Serum Institut, Denmark), Dr. Risto Kayanne (Institute for Molecular Medicine Finland) and the International Head-ache Genetics Consortium for providing GWAS summary statistics for their respective cohorts. Ischemic stroke summary statistics were accessed through the ISGC Cerebrovascular Disease Knowledge Portal. We would like to thank Dr. Weihua Meng (University of Dundee), Dr. Mark Adams (University of Edinburgh) and Dr. Ynte Ruigrok (UMC Utrecht), Dr. Bjarke Feenstra (Statens Serum Institut, Denmark), Dr. Risto Kayanne (Institute for Molecular Medicine Finland), and the International Headache Genetics Consortium for providing GWAS summary statistics for their respective cohorts. We would like to thank the Ming Fund for providing funding for Re. St. M.McC. has received funding from the European Union's Horizon 2020 research and innovation programme under the Marie Skłodowska-Curie grant agreement No 751761. This work was in part supported by an award by a Translational Research Scholars award from the Health Research Board of Ireland (C.D.W.), by research grants from Science Foundation Ireland (SFI) (16/RC/3948 and X) and co-funded under the European Regional Development Fund and by FutureNeuro industry partners. Further funding sources include: Wellcome Trust (grant 084730); Epilepsy Society, UK, NIHR (08-08-SCC); GIHE: NIH R01-NS-49306-01 (R.J.B.); NIH R01-NS-053998 (D.H.L); GSCFE: NIH R01-NS-064154-01 (R.J.B. and Ha.Ha.); NIH: UL1TR001070, Development Fund from The Children's Hospital of Philadelphia (Ha.Ha.); NHMRC Program Grant ID: 1091593 (S.F.B., I.E.S., K.L.O., and K.E.B.); The Royal Melbourne Hospital Foundation Lottery Grant (S.P.); The RMH Neuroscience Foundation (T.J.O'B.); European Union's Seventh Framework Programme (FP7/2007-2013) under grant agreement n° 279062 (EpiPGX) and 602102, Department of Health's NIHR Biomedical Research Centers funding scheme, European Community (EC: FP6 project EPICURE: LSHM-CT-2006-037315); German Research Foundation (DFG: SA434/4-1/4-26-1 (Th.Sa.), WE4896/3-1); EuroEPINOMICS Consortium (European Science Foundation/DFG: SA434/5-1, NU50/8-1, LE1030/11-1, HE5415/3-1 (Th.Sa., P.N., H.L., I.H.), RO 3396/2-1); the German Federal Ministry of Education and Research, National Genome Research Network (NGFNplus/EMINet: 01GS08120, and 01GS08123 (Th.Sa., H.L.); IntenC, TUR 09/I10 (Th.Sa.)); The Netherlands National Epilepsy Fund (grant 04-08); EC (FP7 project EpiPGX 279062). Research Grants Council of the Hong Kong Special Administrative Region, China project numbers HKU7623/08 M (S.S.C, P.K, L.W.B., P.C.S) HKU7747/07 M (S.S.C., P.C.S.) and CUHK4466/06 M (P.K., L.B). Collection of Belgian cases was supported by the Fonds National de la Recherche Scientifique, Fondation Erasme, Université Libre de Bruxelles. GlaxoSmithKline funded the recruitment and data collection for the GenEpA Consortium samples. We acknowledge the support of Nation-wide Children's hospital in Columbus, Ohio, USA. The Wellcome Trust (WT066056) and The NIHR Biomedical Research Centres Scheme (P31753) supported UK contributions. Further support was received through the Intramural Research Program of the Eunice Kennedy Shriver National Institute of Child Health and Human Development (Contract: N01HD33348). The project was also supported by the popgen 2.0 network through a grant from the German Ministry for Education and Research (01EY1103). Parts of the analysis of this work were performed on resources of the High Performance Center of the University of Luxembourg and Elixir-Luxembourg. The KORA study was initiated and financed by the Helmholtz Zentrum München – German Research Center for Environmental Health, which is funded by the German Federal Ministry of Education and Research (BMBF) and by the State of Bavaria. Furthermore, KORA research was supported within the Munich Center of Health Sciences (MC-Health), Ludwig-Maximilians-Universität, as part of LMUinnovativ. The International League Against Epilepsy (ILAE) facilitated the Consortium through the Commission on Genetics and by financial support; however, the opinions expressed in the manuscript do not necessarily represent the policy or position of the ILAE.

## Author contributions

Data analysis: protocol development and main analyses: G.L.C., B.P.C.K., Ro.Kr. (data management), De.La., C.L., M.McC. (co-lead analyst), N.M., D.S., and Re.St. (co-lead analyst); analysis coordination: G.L.C., B.P.C.K., and D.S.; data preparation, imputation and quality control: M.B., D.J.B., L.B., J.P.B., R.J.B., G.L.C., S.S.C., A.J.C., C.G.F.deK., S.F., D.B.G., H.G., Y.G., Ha.Ha., E.L.H., I.H., A.I., D.K., B.P.C.K., Ro.Kr., De.La., C.L., I.L-C., A.M., M.McC., N.M., P.-W.N., P.N., Sa.Pe., Sl.Pe., Th.Sa., P.C.S., A.S., D.S., Re.St., Z.W., C.D.W., and Fe.Za.; analysis review: D.J.B., and Ha.Ha. Writing committee: S.F.B., G.L.C., B.P.C.K., M.McC. (co-wrote first draft), and Re.St. (co-wrote first draft). Strategy committee: L.B., S.F.B., R.J.B., G.L.C., Ha.Ha., E.L.H., M.R.J., Re.Kä., B.P.C.K., Ro.Kr., P.K., H.L., I.L-C., T.J.O'B., and S.M.S. Phenotyping committee: C.D., D.J.D., W.S.K., P.K., D.H.L., A.G.M., M.R.S., and P.S. Governance committee: S.F.B., Al.Co., A.-E.L., and D.H.L. Patient recruitment and phenotyping: B.A.-K., P.A., A.A., T.B., A.J.B., F.B., B.B., S.F.B., R.J.B., E.C., G.D.C., C.B.C., K.C., An.Co., P.C., J.J.C., G-J.deH., P.De.J., N.D., C.D., O.D., D.J.D., C.P.D., C.E.E., T.N.F., M.F., B.F., J.A.F., V.G., E.B.G., T.G., S.G., K.F.H., K.H., S.H., I.H., C.H., He.Hj., M.I., J.J-K., M.R.J., Re.Kä., A.-M.K., D.K.-N.T., H.E.K., R.C.K., M.K., W.S.K., Ru.Ku., P.K., H.L., Di.Li., W.D.L., I.L-C., D.H.L., A.G.M., T.M., M.M., R.S.M., H.M., M.N., P.-W.N., T.J.O'B., An.Po., M.P., R.R., S.R., P.S.R., E.M.R., F.R., J.W.S., Th.Sa., Th.Sc., S.C.S., C.J.S., I.E.S., B.S., S.S., J.J.S., G.J.S., S.M.S., L.S., D.F.S., M.C.S., P.E.S., A.C. M.S., M.R.S., B.J.S., U.S., P.S., H.S., Ra.Su., K.M.T., L.L.T., M.T., R.T., M.S.V., E.P.G.V., F.V., S.v.S., N.M.W., Y.G.W., J.W., C.D.W., P.W.-W., M.W., S.W., and Fr.Zi. Control

cohorts: L.C.B., J.G.E., A.F., C.G., Ha.Ha., Y.-L.L., Wo.Li., J.L.M., A.M.M., M.M.N., Aa. Pa., F.P., K.S., H.S., G.N.T., and W.Y. Consortium coordination: K.E.B. and K.L.O.

## Additional information

**Competing interests:** The authors declare no competing interests.

## The International League Against Epilepsy Consortium on Complex Epilepsies

Bassel Abou-Khalil[1], Pauls Auce[2,3], Andreja Avbersek[4], Melanie Bahlo[5,6,7], David J. Balding[8,9], Thomas Bast[10,11], Larry Baum[12], Albert J. Becker[13], Felicitas Becker[14,15], Bianca Berghuis[16], Samuel F. Berkovic[17], Katja E. Boysen[17], Jonathan P. Bradfield[18,19], Lawrence C. Brody[20], Russell J. Buono[18,21,22], Ellen Campbell[23], Gregory D. Cascino[24], Claudia B. Catarino[4], Gianpiero L. Cavalleri[25,26], Stacey S. Cherny[27,28], Krishna Chinthapalli[4], Alison J. Coffey[29], Alastair Compston[30], Antonietta Coppola[31,32], Patrick Cossette[33], John J. Craig[34], Gerrit-Jan de Haan[35], Peter De Jonghe[36,37], Carolien G.F. de Kovel[38], Norman Delanty[25,26,39], Chantal Depondt[40], Orrin Devinsky[41], Dennis J. Dlugos[42], Colin P. Doherty[26,43], Christian E. Elger[44], Johan G. Eriksson[45], Thomas N. Ferraro[21,46], Martha Feucht[47], Ben Francis[48], Andre Franke[49], Jacqueline A. French[50], Saskia Freytag[5], Verena Gaus[51], Eric B. Geller[52], Christian Gieger[53,54], Tracy Glauser[55], Simon Glynn[56], David B. Goldstein[57,58], Hongsheng Gui[27], Youling Guo[27], Kevin F. Haas[1], Hakon Hakonarson[18,59], Kerstin Hallmann[44,60], Sheryl Haut[61], Erin L. Heinzen[57,58], Ingo Helbig[42,62], Christian Hengsbach[14], Helle Hjalgrim[63,64], Michele Iacomino[32], Andrés Ingason[65], Jennifer Jamnadas-Khoda[4,66], Michael R. Johnson[67], Reetta Kälviäinen[68,69], Anne-Mari Kantanen[68], Dalia Kasperavičiūte[4], Dorothee Kasteleijn-Nolst Trenite[38], Heidi E. Kirsch[70], Robert C. Knowlton[71], Bobby P.C. Koeleman[38], Roland Krause[72], Martin Krenn[73], Wolfram S. Kunz[44], Ruben Kuzn200iecky[74], Patrick Kwan[12,75,76], Dennis Lal[77], Yu-Lung Lau[78], Anna-Elina Lehesjoki[79], Holger Lerche[14], Costin Leu[4,77,80], Wolfgang Lieb[81], Dick Lindhout[35,38], Warren D. Lo[82], Iscia Lopes-Cendes[83,84], Daniel H. Lowenstein[70], Alberto Malovini[85], Anthony G. Marson[2], Thomas Mayer[86], Mark McCormack[25], James L. Mills[87], Nasir Mirza[2], Martina Moerzinger[47], Rikke S. Møller[63,64,88], Anne M. Molloy[89], Hiltrud Muhle[62], Mark Newton[90], Ping-Wing Ng[91], Markus M. Nöthen[92], Peter Nürnberg[93], Terence J. O'Brien[75,76], Karen L. Oliver[17], Aarno Palotie[94,95], Faith Pangilinan[20], Sarah Peter[72], Slavé Petrovski[75,96], Annapurna Poduri[97], Michael Privitera[98], Rodney Radtke[99], Sarah Rau[14], Philipp S. Reif[100,101], Eva M. Reinthaler[73], Felix Rosenow[100,101], Josemir W. Sander[4,35,102], Thomas Sander[51,93], Theresa Scattergood[103], Steven C. Schachter[104], Christoph J. Schankin[105], Ingrid E. Scheffer[17,106], Bettina Schmitz[51], Susanne Schoch[13], Pak C. Sham[27], Jerry J. Shih[107], Graeme J. Sills[2], Sanjay M. Sisodiya[4,102], Lisa Slattery[108], Alexander Smith[77], David F. Smith[3], Michael C. Smith[109], Philip E. Smith[110], Anja C.M. Sonsma[38], Doug Speed[8,111], Michael R. Sperling[112], Bernhard J. Steinhoff[10], Ulrich Stephani[62], Remi Stevelink[38], Konstantin Strauch[113,114], Pasquale Striano[115], Hans Stroink[116], Rainer Surges[44], K. Meng Tan[75], Liu Lin Thio[117], G. Neil Thomas[118], Marian Todaro[75], Rossana Tozzi[119], Maria S. Vari[115], Eileen P.G. Vining[120], Frank Visscher[121], Sarah von Spiczak[62], Nicole M. Walley[57,122], Yvonne G. Weber[14], Zhi Wei[123], Judith Weisenberg[117], Christopher D. Whelan[25],

Peter Widdess-Walsh[52], Markus Wolff[124], Stefan Wolking[14], Wanling Yang[78], Federico Zara[32] & Fritz Zimprich[73]

[1]Vanderbilt University Medical Center, Nashville, TN 37232, USA. [2]Department of Molecular and Clinical Pharmacology, University of Liverpool, Liverpool L69 3GL, UK. [3]The Walton Centre NHS Foundation Trust, Liverpool L9 7LJ, UK. [4]Department of Clinical and Experimental Epilepsy, UCL Institute of Neurology, Queen Square, London WC1N 3BG, UK. [5]Population Health and Immunity Divison, The Walter and Eliza Hall Institute of Medical Research, Parkville 3052, Australia. [6]Department of Biology, University of Melbourne, Parkville 3010, Australia. [7]School of Mathematics and Statistics, University of Melbourne, Parkville 3010, Australia. [8]UCL Genetics Institute, University College London, London WC1E 6BT, UK. [9]Melbourne Integrative Genomics, University of Melbourne, Parkville 3052, Australia. [10]Epilepsy Center Kork, Kehl-Kork 77694, Germany. [11]Medical Faculty of the University of Freiburg, Freiburg 79085, Germany. [12]Centre for Genomic Sciences, The University of Hong Kong, Hong Kong, Hong Kong. [13]Section for Translational Epilepsy Research, Department of Neuropathology, University of Bonn Medical Center, Bonn 53105, Germany. [14]Department of Neurology and Epileptology, Hertie Institute for Clinical Brain Research, University of Tübingen, Tübingen 72076, Germany. [15]Department of Neurology, University of Ulm, Ulm 89081, Germany. [16]Stichting Epilepsie Instellingen Nederland (SEIN), Zwolle 8025 BV, The Netherlands. [17]Epilepsy Research Centre, University of Melbourne, Austin Health, Heidelberg 3084, Australia. [18]Center for Applied Genomics, The Children's Hospital of Philadelphia, Philadelphia, PA 19104, USA. [19]Quantinuum Research LLC, San Diego, CA 92101, USA. [20]National Human Genome Research Institute, National Institutes of Health, Bethesda, MD 20892, USA. [21]Department of Biomedical Sciences, Cooper Medical School of Rowan University Camden, Camden, NJ 08103, USA. [22]Department of Neurology, Thomas Jefferson University Hospital, Philadelphia, PA 19107, USA. [23]Belfast Health and Social Care Trust, Belfast BT9 7AB, UK. [24]Division of Epilepsy, Department of Neurology, Mayo Clinic, Rochester, MN 55902, USA. [25]Department of Molecular and Cellular Therapeutics, The Royal College of Surgeons in Ireland, Dublin 2, Ireland. [26]The FutureNeuro Research Centre, Dublin 2, Ireland. [27]Department of Psychiatry, The University of Hong Kong, Hong Kong, Hong Kong. [28]Department of Epidemiology and Preventive Medicine, School of Public Health, Sackler Faculty of Medicine, Tel Aviv University, Tel Aviv 6997801, Israel. [29]The Wellcome Trust Sanger Institute, Hinxton, Cambridge CB10 1SA, UK. [30]Department of Clinical Neurosciences, Cambridge Biomedical Campus, Cambridge CB2 0SL, UK. [31]Department of Neuroscience, Reproductive and Odontostomatological Sciences, University Federico II, Naples 80138, Italy. [32]Laboratory of Neurogenetics and Neurosciences, Institute G. Gaslini, Genova 16148, Italy. [33]Department of Neurosciences, University of Montreal, Montreal CA 26758, Canada. [34]Department of Neurology, Royal Victoria Hospital, Belfast Health and Social Care Trust, Grosvenor Road, Belfast BT12 6BA, UK. [35]Stichting Epilepsie Instellingen Nederland (SEIN), Heemstede 2103 SW, The Netherlands. [36]Neurogenetics Group, Center for Molecular Neurology, VIB and Laboratory of Neurogenetics, Institute Born-Bunge, University of Antwerp, Antwerp 2610, Belgium. [37]Department of Neurology, Antwerp University Hospital, Edegem 2650, Belgium. [38]Department of Genetics, University Medical Center Utrecht, Utrecht 3584 CX, The Netherlands. [39]Division of Neurology, Beaumont Hospital, Dublin D09 FT51, Ireland. [40]Department of Neurology, Hôpital Erasme, Université Libre de Bruxelles, Brussels 1070, Belgium. [41]Comprehensive Epilepsy Center, New York University School of Medicine, New York, NY 10016, USA. [42]Department of Neurology, The Children's Hospital of Philadelphia, Philadelphia, PA 19104, USA. [43]Neurology Department, St. James's Hospital, Dublin D03 VX82, Ireland. [44]Department of Epileptology, University of Bonn Medical Centre, Bonn 53127, Germany. [45]Department of General Practice and Primary Health Care, University of Helsinki and Helsinki University Hospital, Helsinki 0014, Finland. [46]Department of Pharmacology and Psychiatry, University of Pennsylvania Perlman School of Medicine, Philadelphia, PA 19104, USA. [47]Department of Pediatrics and Neonatology, Medical University of Vienna, Vienna 1090, Austria. [48]Department of Biostatistics, University of Liverpool, Liverpool L69 3GL, UK. [49]Institute of Clinical Molecular Biology, Christian-Albrechts-University of Kiel, University Hospital Schleswig Holstein, Kiel 24105, Germany. [50]Department of Neurology, NYU School of Medicine, New York City, NY 10003, USA. [51]Department of Neurology, Charité Universitaetsmedizin Berlin, Campus Virchow-Clinic, Berlin 13353, Germany. [52]Institute of Neurology and Neurosurgery at St. Barnabas, Livingston, NJ 07039, USA. [53]Research Unit of Molecular Epidemiology, Helmholtz Zentrum München - German Research Center for Environmental Health, Neuherberg D-85764, Germany. [54]Institute of Epidemiology, Helmholtz Zentrum München - German Research Center for Environmental Health, Neuherberg D-85764, Germany. [55]Comprehensive Epilepsy Center, Division of Neurology, Cincinnati Children's Hospital Medical Center, Cincinnati, OH 45229, USA. [56]Department of Neurology, University of Michigan, Ann Arbor, MI 48109, USA. [57]Center for Human Genome Variation, Duke University School of Medicine, Durham, NC 27710, USA. [58]Institute for Genomic Medicine, Columbia University Medical Center, New York, NY 10032, USA. [59]Division of Human Genetics, Department of Pediatrics, The Perelman School of Medicine, University of Pennsylvania, Philadelphia, PA 19104, USA. [60]Life and Brain Center, University of Bonn Medical Center, Bonn 53127, Germany. [61]Montefiore Medical Center, Bronx, NY 10467, USA. [62]Department of Neuropediatrics, University Medical Center Schleswig-Holstein (UKSH), Kiel 24105, Germany. [63]Danish Epilepsy Centre, Dianalund 4293, Denmark. [64]Institute of Regional Health Services Research, University of Southern Denmark, Odense 5000, Denmark. [65]deCODE genetics, Inc., Reykjavik IS-101, Iceland. [66]Department of Psychiatry and Applied Psychology, Institute of Mental Health University of Nottingham, Nottingham NG7 2TU, UK. [67]Faculty of Medicine, Imperial College London, London SW7 2AZ, UK. [68]Kuopio Epilepsy Center, Neurocenter, Kuopio University Hospital, Kuopio 70029, Finland. [69]Institute of Clinical Medicine, University of Eastern Finland, Kuopio 70029, Finland. [70]Department of Neurology, University of California, San Francisco, CA 94143, USA. [71]University of Alabama Birmingham, Department of Neurology, Birmingham, AL 35233, USA. [72]Luxembourg Centre for Systems Biomedicine, University of Luxembourg, Esch-sur-Alzette L-4362, Luxembourg. [73]Department of Neurology, Medical University of Vienna, Vienna 1090, Austria. [74]Department of Neurology, Zucker-Hofstra Northwell School of Medicine, New York, NY 10075, USA. [75]Department of Medicine, University of Melbourne, Royal Melbourne Hospital, Parkville, VIC 3050, Australia. [76]Department of Neuroscience, Central Clinical School, Monash University, Melbourne, VIC 3004, Australia. [77]Stanley Center for Psychiatric Research, Broad Institute of Harvard and M.I.T, Cambridge, MA 02142, USA. [78]Department of Paediatrics and Adolescent Medicine, The University of Hong Kong, Hong Kong, Hong Kong. [79]Folkhälsan Research Center and Medical Faculty, University of Helsinki, Helsinki 00290, Finland. [80]Genomic Medicine Institute, Lerner Research Institute, Cleveland Clinic, Cleveland, OH 44195, USA. [81]Institut für Epidemiologie Christian-Albrechts-Universität zu Kiel, Kiel 24105, Germany. [82]Department of Pediatrics and Neurology, Ohio State University and Nationwide Children's Hospital, Columbus, OH 43205, USA. [83]Department of Medical Genetics, School of Medical Sciences, University of Campinas (UNICAMP), Campinas 13083-887 SP, Brazil. [84]Brazilian Institute of Neuroscience and Neurotechnology (BRAINN), Campinas, SP 13083-970, Brazil. [85]Istituti Clinici Scientifici Maugeri, Pavia 27100, Italy. [86]Epilepsy Center Kleinwachau, Radeberg 01454, Germany. [87]Division of Intramural Population Health Research, Eunice Kennedy Shriver National Institute of Child Health and Human Development, National Institutes of Health, Bethesda, MD 20892, USA. [88]Wilhelm Johannsen Centre for Functional Genome Research, Copenhagen DK-2200, Denmark. [89]School of Medicine, Trinity College Dublin, Dublin 2, Ireland. [90]Department of Neurology, Austin Health, Heidelberg, VIC 3084, Australia. [91]United Christian Hospital, Hong Kong, Hong Kong. [92]Institute of Human Genetics, University of Bonn Medical Center, Bonn 53127, Germany. [93]Cologne Center for Genomics, University of Cologne, Cologne 50931, Germany. [94]Institute for Molecular Medicine Finland (FIMM), University of Helsinki, Helsinki 0014, Finland. [95]The Broad Institute of M.I.T. and Harvard, Cambridge, MA 02142, USA. [96]AstraZeneca Centre for Genomics Research, Precision Medicine and Genomics, IMED Biotech Unit, AstraZeneca, Cambridge CB2 0AA, UK. [97]Department of Neurology, Boston Children's Hospital,

Harvard Medical School, Boston, MA 02115, USA. [98]Department of Neurology, Neuroscience Institute, University of Cincinnati Medical Center, Cincinnati, OH 45220, USA. [99]Department of Neurology, Duke University School of Medicine, Durham, NC 27710, USA. [100]Epilepsy-Center Hessen, Department of Neurology, University Medical Center Giessen and Marburg, Marburg, Germany and Philipps-University Marburg, Marburg 35043, Germany. [101]Epilepsy Center Frankfurt Rhine-Main, Center of Neurology and Neurosurgery, Goethe University Frankfurt, Frankfurt 60528, Germany. [102]Chalfont Centre for Epilepsy, Chalfont-St-Peter, Buckinghamshire SL9 0RJ, UK. [103]Department of Endocrinology, Hospital of The University of Pennsylvania, Philadelphia, PA 19104, USA. [104]Departments of Neurology, Beth Israel Deaconess Medical Center, Massachusetts General Hospital, and Harvard Medical School, Boston, MA 02215, USA. [105]Department of Neurology, Inselspital, Bern University Hospital, University of Bern, Bern 3010, Switzerland. [106]Department of Neurology, Royal Children's Hospital, Parkville, VIC 3052, Australia. [107]Department of Neurosciences, University of California, San Diego, La Jolla, CA 92037, USA. [108]The Royal College of Surgeons in Ireland, Dublin D02 YN77, Ireland. [109]Rush University Medical Center, Chicago 60612 IL, USA. [110]Department of Neurology, Alan Richens Epilepsy Unit, University Hospital of Wales, Cardiff CF14 4XW, UK. [111]Aarhus Institute of Advanced Studies (AIAS), Aarhus University, Aarhus 8000, Denmark. [112]Department of Neurology and Comprehensive Epilepsy Center, Thomas Jefferson University, Philadelphia, PA 19107, USA. [113]Institute of Genetic Epidemiology, Helmholtz Zentrum München - German Research Center for Environmental Health, Neuherberg, Neuherberg D-85764, Germany. [114]IBE, Faculty of Medicine, LMU Munich, Munich 80539, Germany. [115]Pediatric Neurology and Muscular Diseases Unit, Department of Neurosciences, Rehabilitation, Ophthalmology, Genetics, Maternal and Child Health, G. Gaslini Institute, University of Genoa, Genova 16148, Italy. [116]CWZ Hospital, Nijmegen 6532 SZ, The Netherlands. [117]Department of Neurology, Washington University School of Medicine, St. Louis, MO 63110, USA. [118]Institute for Applied Health Research, University of Birmingham, Birmingham B15 2TT, UK. [119]C. Mondino National Neurological Institute, Pavia 27100, Italy. [120]Departments of Neurology and Pediatrics, The Johns Hopkins University School of Medicine, Baltimore, MD 21287, USA. [121]Department of Neurology, Admiraal De Ruyter Hospital, Goes 4462, The Netherlands. [122]Division of Medical Genetics, Department of Pediatrics, Duke University Medical Center, Durham, NC 27710, USA. [123]Department of Computer Science, New Jersey Institute of Technology, New Jersey, NJ 07102, USA. [124]Department of Pediatric Neurology and Developmental Medicine, University Children's Hospital, Tübingen 72076, Germany

