## [Peer Review File · Nature Communications]

Reviewer #1 (Remarks to the Author):

The authors performed a GWAS with 15,212 cases of epilepsy (3 broad types and 7 subtypes) and 29,677 controls. Overall, the GWAS procedure and statistics appear sound. The only issue was some apparent genomic inflation, for which the authors seem to give a reasonable explanation. The authors found 16 significant loci of which 11 are novel.

They next used functional data including TWAS, eQTL, gene expression in brain, missense mutations, protein interactions, and knockout mouse phenotypes to implicate genes near the loci. This part was interesting, however equally weighing these data and scoring them to determine the plausibility of genes being involved in epilepsy seemed superficial, as there are all different types of functional evidence, and not all of them (e.g. the PPI) are from brain. A more data-driven way to implicate epilepsy genes with disparate data types may be to take a comprehensive list of rare variants with epilepsy as a known phenotype and assess which of these evidence sources reliably implicates the known genes in epilepsy, if this is possible. Another way that is more biologically relevant might be to identify if the variants are in promoters or enhancers and use Hi-C data to implicate specific genes, as has been done by some recent studies (PMID: 29632383, 27760116). The authors should at least try this approach, though they might not find much based on their chromatin mark enrichment in Figure 2.

The authors then go on to perform heritability and genetic correlation analyses, where they find that the heritability for generalized epilepsy is moderate ($h^2 > 0.3$) while it is quite low for focal epilepsies. Additionally, they find that focal epilepsies appear not to have similarity with the generalized epilepsies, which is intuitive but quite interesting to see borne out with common variation. I wonder if they might find that the genetic signal is more confined to genes expressed in early brain development for the generalized epilepsies compared to focal epilepsies.

The authors also assess genetic correlations for each epilepsy type with different diseases. These results were less straightforward to reconcile with known epidemiology and neurobiology. I was surprised the only significant association was with cognitive ability. I would have expected a stronger association between autism and epilepsy, given that about 30-50% of individuals with autism have epilepsy. In some genes, e.g. SCN2A, there is evidence that rare damaging mutations are associated with more severe phenotypes such as autism or intellectual disability, while less damaging missense mutations may be associated with less severe phenotypes like epilepsy (PMID: 25163687). I also wonder if common variants may be enriched in the promoters/introns/regulatory elements of genes where there are rare variants in autism or intellectual disability. I also wonder if there is any relationship to the genetic signal from febrile seizures, where there are a few genome wide hits (PMID: 25344690).

Finally, the use of the LINCS data to identify drugs was interesting, but not followed up on in any meaningful way. It would be nice to at least see some validation that some of the top drugs, when used in a mouse model, result in the expected gene expression changes. I am sure there are a few drugs out of the top 20 or 30 that have been tested on mice with gene expression data from brain.

Major issues

-Please clarify upfront which results are from a mega-analysis vs meta-analysis, and whether there is any bona fide replication in this study.

-Implicating genes with promoters/enhancers and Hi-C and/or eQTL/TWAS data only would be much more convincing than the equal weighting of six arbitrary data types and tallying up scores approach used here

-Please see my comments about genetic correlations and rare variants above. I think it would be worthwhile to:

--Assess the co-heritability of febrile seizures using the existing GWAS (there is a known epidemiological risk association) and perhaps GWAS for other disorders that are known to be co-morbid with genetic epilepsies

--Assess enrichment with MAGMA or stratified LD score regression (PMID 26414678) with genes or the promoters/enhancers of genes harboring rare variants in epilepsy and ASD.

These analyses would test biological hypothesis about epilepsy comorbidities and re-assure me that Figure 4 is accurately computed.

Minor points

-It would be informative to see whether there is differential enrichment in genes expressed prenatally vs postnatally vs in later life across the epilepsies.

-Mouse model validation with existing data for some of the drugs would add excellent support for the otherwise highly speculative connectivity map analysis, as mentioned above.

-Figure 2D and S10 don't have labels on the x-axis. I would cut down the number of bars and just show the top 10, with labels underneath.

-For the enrichment analysis for common variants in monogenic epilepsy genes and antiepileptic drugs: background size, odds ratio, and rationale for background sets would be informative. It is known that longer genes are more highly expressed in brain and this can bias results, etc.

-It would be helpful to discuss if future studies will be on refractory vs non-refractory epilepsy, or if there might be a way to study drug responsiveness in epilepsies

Overall, this is a very interesting and valuable study that further characterizes the contribution of common variation to different subtypes of epilepsy. The functional analyses with the GWAS data were very extensive, though not coherent and had some results that were counterintuitive and were not well explained. It would be helpful to have additional analyses that support the weaker analyses and some additional discussion about these results.

Reviewer #2 (Remarks to the Author):

This is a well conducted and clearly presented investigation into the genetic bases of the epilepsies.

They have a large series of patients >15K and almost twice as many controls.

They have categorised the patient into one of 3 groups- focal epilepsies, genetic generalised and unclassified. As part of the further analyses they have also investigated the role of the identified loci in some of the well recognised sub-classifications- eg JME, CAE etc.

The samples of been genotyped using widely available arrays and appropriate QC has been applied.

The history of this sort of broad approach of using GWAS in the epilepsies was largely little disappointing in the early years- took small or underpowered studies. This despite there being high expectation, due the heritability estimates, that GWAS ought to offer a successful approach. The first chinks appeared a few years ago with the largest scale (at that time) and now this. to this reviewers knowledge the largest, most comprehensive study so far. They report several new loci and confirm 3 of the previously identified loci.

I think it woill be important for them to specify when compared to the previous gWAS by this group involving 8696 cases and this one with 15.2K cases if there is any overall;p ie are there 7.5K new cast or are these a completely new set.

This will help the reader know how much weight to put by the confirmation of the older loci.

I also think a little note one caution needs to be struck as these findings have not yet been replicated in a further separately acquired cohort.

I think particular caution in interpretation in the sub-group analyses- ie when a gene is being reported as significant in a clinical sub-type. It would b very reassuring if a second similarly phenotyped cohort could be analysed. I appreciate however that it is very unlikely that this consortium has available another large series of patients but I do think the need to reflect these issues in their paper.

The second part of their manuscript deal with a variety of approaches including TWAS etc to try and prioritise the underlying gene as opposed to just describing the loci. I think this does add value and although there are no current absolute 'gold' standards to prove a gene they have used a variety of tools and analyses to bolster the evidence behind the genes named. Whether all of these genes will in time (and replication) prove to be the gene remains to be seen.

It is also intriguing to note, as the authors discuss, that a significant proportion of these prioritised genes are known drug targets supporting the view that maybe some of the eaters provide putative targets.

Overall I though this was an impressive piece of work.

Reviewer #3 (Remarks to the Author):

The ILAE consortium performed the largest GWAS on epilepsy, almost doubling the number of cases in their study, and identify 11 novel loci. The manuscript is rich in follow-up analyses, but lacks in depth discussion of the clinical and biological implications of their results. With proper revisions, this manuscript will be well suited for the journal.

Overall:

1. Understandably, the paper is transferred and formatted as a brief communication, but a revised version should make use of the available space and increase the depth and clarity (see other points below).

Methods:

2. While this analysis increases the number of cases to 15k, one wonders whether the researchers were not able to incorporate additional data from large biobank such as the UKB. Was epilepsy simply not ascertained in any other dataset?

3. Hypergeometric test: to what extent were other factors taken into account for the enrichment analyses (e.g., gene size, location)? when comparing to all protein-coding genes, it is not unreasonable to imagine GWAS hits (regulatory variants) in general may enrich specific gene sets. Did the authors also perform this analysis (enrichment for monogenic genes) for GWAS hits of other traits, e.g. height / diabetes / alzheimers?

4. Genetic correlation: did the authors consider neuroimaging traits? hippocampal volume, icv, other subcortical structures.

Results:

5. The manhattan plot in figure 1 could be made more informative by marking the novel loci and perhaps gene/snp names.

6. I miss a table/figure/heatmap in the main text showing the associations of all 11(/16) loci with all subtypes. Perhaps this could even replace some of the text in the results section.

Discussion:

7. Perhaps if/when the authors reformat their paper, a more thorough discussion could be added. What does this major advance mean for the? does it point to subtypes that could merged, or redefining diagnoses altogether give the lack of heritability? Or are these subtypes real but simply environmentally driven. Could these findings help with diagnosis making, perhaps prognosis/prediction? The biological discussion does go a bit further at times, but it would be nice if the authors can leverage the extra space they now have.

Genome-wide mega-analysis identifies 11 new loci and highlights diverse biological mechanisms in the common epilepsies

The International League Against Epilepsy Consortium on Complex Epilepsies

Response to Reviewers

We thank the editor for allowing us to revise our manuscript and the reviewers for their constructive comments, which have now been addressed. We believe the manuscript has improved as a result.

We include the complete text of each reviewer shown in *italics*. Our response to each individual comment is detailed below with specific text changes in **red**.

Reviewer #1

The authors performed a GWAS with 15,212 cases of epilepsy (3 broad types and 7 subtypes) and 29,677 controls. Overall, the GWAS procedure and statistics appear sound. The only issue was some apparent genomic inflation, for which the authors seem to give a reasonable explanation. The authors found 16 significant loci of which 11 are novel.

They next used functional data including TWAS, eQTL, gene expression in brain, missense mutations, protein interactions, and knockout mouse phenotypes to implicate genes near the loci. This part was interesting, however equally weighing these data and scoring them to determine the plausibility of genes being involved in epilepsy seemed superficial, as there are all different types of functional evidence, and not all of them (e.g. the PPI) are from brain. A more data-driven way to implicate epilepsy genes with disparate data types may be to take a comprehensive list of rare variants with epilepsy as a known phenotype and assess which of these evidence sources reliably implicates the known genes in epilepsy, if this is possible.

Response 1.1

In order to address this important point, we have calculated the correlation, sensitivity and specificity of these 6 criteria in predicting established monogenic epilepsy genes within our data (*PNPO, SCN1A, SCN2A, SCN9A, STX1B, ZEB2*) to validate our gene prioritization method. As seen in the below table (now included as **Supplementary Table 3, and given below**), individually, each criterion on its own has limited predictive value. However, in combination, or

as a sum, they are strong predictors of established epilepsy genes. Moreover, our current approach of selecting the gene(s) with the highest total score within a locus yields a sensitivity of 83% and a specificity of 89% to detect established epilepsy genes, which is unlikely to improve by implementing weights.

We agree the approach we employed of weighting all criteria equally (adapted from Nature PMID:24390342), might seem somewhat arbitrary. However, given the results of this validation check described above, we feel that the low sensitivity of the TWAS and eQTL criteria would make them unsuitable for use as sole prioritization criteria. Furthermore, there are loci without a single eQTL or TWAS association, and others with multiple eQTL/TWAS associations; it is the latter case in particular where we feel the addition of the other 4 criteria facilitate a rational and data-driven approach to prioritization.

	TWAS	eQTL	Brain exp e	Missens	PPI	KO mouse	Total score
Correlation coef	0.07	0.08	0.33	0.13	0.14	0.26	0.42
Sensitivity	0.17	0.17	1.00	0.17	0.50	0.83	0.83
Specificity	0.93	0.94	0.74	0.96	0.79	0.75	0.89
Postive predictive value	0.09	0.10	0.14	0.17	0.09	0.13	0.24
Negative predictive value	0.96	0.96	1.00	0.96	0.97	0.99	0.99

Supplementary Table 3: Validation of the 6 biological prioritization criteria (**Table 1**) to predict established monogenic epilepsy genes. A list of 102 established monogenic epilepsy genes (see **Supplementary Table 10**) was used to calculate the correlation, sensitivity, specificity, positive predictive value and negative predictive value of the prioritization criteria in predicting established epilepsy genes out of the 146 genes that were mapped to genome-wide significant loci (**Supplementary Table 2**). The total score is the sum of criteria being met per gene (range 0-6). Correlation coefficients are calculated as Pearson correlations. The “total score” column represents our approach of selecting the gene(s) with the highest score within each locus as prioritized epilepsy genes.

Another way that is more biologically relevant might be to identify if the variants are in promoters or enhancers and use Hi-C data to implicate specific genes, as has been done by some recent studies (PMID: 29632383, 27760116). The authors should at least try this approach, though they might not find much based on their chromatin mark enrichment in Figure 2.

Response 1.2

We point the reviewer towards Methods section ‘Gene mapping and biological prioritization’, where we describe how we used Hi-C data to map epilepsy loci emerging from our GWAS, to distal genes that show a significant 3D chromatin interaction with a gene promoter (FDR < 10⁻⁶). See Methods section “Gene mapping and biological prioritization” line 5, which now reads: **Additionally, we mapped genes that were farther than 250kb away from the locus using chromatin interaction data to identify genes that show a significant 3D interaction (P_{FDR}<10⁻⁶)**

between their promoter and the locus, based on Hi-C data from dorsolateral prefrontal cortex, hippocampus and neural progenitor cells.⁵⁴).

However, we have not used Hi-C data as a prioritization criterion, since few loci have a significant Hi-C interaction and we have no evidence to support that the distal Hi-C mapped genes are more likely to be causal than the genes that are located within the significant locus.

The authors then go on to perform heritability and genetic correlation analyses, where they find that the heritability for generalized epilepsy is moderate ($h^2 > 0.3$) while it is quite low for focal epilepsies. Additionally, they find that focal epilepsies appear not to have similarity with the generalized epilepsies, which is intuitive but quite interesting to see borne out with common variation. I wonder if they might find that the genetic signal is more confined to genes expressed in early brain development for the generalized epilepsies compared to generalized epilepsies.

Response 1.3

We thank the reviewer for this insightful suggestion. We have now applied our data to the Brainspan database, as implemented in FUMA, to assess whether the genes implicated by our GWAS are differentially expressed in the brain at various prenatal and post-natal ages. We have performed these analyses for the genes prioritized in any epilepsy phenotype (21 genes), any focal epilepsy phenotype (8 genes) and any generalized epilepsy phenotype (17 genes). The results, summarized in the Figure below, suggests that expression of the genes associated with focal epilepsies is up-regulated in late-infancy and young adulthood, whereas expression of those genes associated with generalized epilepsy is down-regulated in early childhood and differentially expressed prenatally and at adolescence.

We believe this interesting information fits well in the context of our current study and have added text (describing these results) to the section entitled “SNP annotation and tissue specific partitioned heritability analyses”, pg 6, and added the figure as **Supplemental Figure 13**. The text included is as follows: **Finally, we leveraged the the Brainspan database, as implemented in FUMA, to assess whether the genes implicated by our GWAS are differentially expressed in the brain at various prenatal and post-natal ages. These analyses were performed for the genes prioritized in any epilepsy phenotype (21 genes), any focal epilepsy phenotype (8 genes) or any generalized epilepsy phenotype (17 genes). The results suggest that expression of the genes associated with focal epilepsies is up-regulated in late-infancy and young adulthood, whereas expression of those genes associated with generalized epilepsy is down-regulated in early childhood and differentially expressed prenatally and at adolescence (Supplement Figure 13).**

Supplementary Figure 13: Differentially expressed genes across the lifespan for all (A), focal (B) and generalized epilepsies (C). Significantly enriched DEG sets ($P_{\text{bon}} < 0.05$) are highlighted in red. Gene transcription data from postmortem human brain specimens from the Brainspan Atlas of the Developing Human Brain was used.

The authors also assess genetic correlations for each epilepsy type with different diseases. These results were less straightforward to reconcile with known epidemiology and neurobiology. I was surprised the only significant association was with cognitive ability. I would have expected a stronger association between autism and epilepsy, given that about 30-50% of individuals with autism have epilepsy. In some genes, e.g. SCN2A, there is evidence that rare damaging mutations are associated with more severe phenotypes such as autism or intellectual disability, while less damaging missense mutations may be associated with less severe phenotypes like epilepsy (PMID: 25163687). I also wonder if common variants may be enriched in the promoters/introns/regulatory elements of genes where there are rare variants in autism or intellectual disability.

Response 1.4

We agree with the reviewer that an assessment of co-heritability of epilepsy with other co-morbid disorders is interesting and worthwhile. A co-heritability assessment (calculated using LD-score regression and expressed as genetic correlation) of epilepsy with febrile seizures and several other disorders which are often co-morbid with epilepsy is presented in **Figure 4** and **Supplementary Table 7**.

We too were initially surprised by the lack of genetic correlation between ASD and epilepsy. However, our findings are consistent with a recent publication in *Science* by the Brainstorm Consortium (PMID 29930110) which, using summary statistics from our first epilepsy GWAS (Lancet Neurology, 2014), found a negative correlation between ASD and epilepsy. More importantly, the Brainstorm study showed neurological diseases to be relatively distinct from each other at a genetic level and did not find strong correlations between neurological and psychiatric diseases.

To further investigate the genetic correlation between ASD and epilepsy, and following the reviewer suggestion of investigating whether our GWAS is enriched for genes harbouring rare variants in epilepsy and ASD, we attained a list of monogenic ASD genes from the SFARI database (<https://gene.sfari.org/>), from which we extracted the 83 genes with a score of 2 (strong candidate gene) or 1 (high confidence candidate gene).

Next, we assessed whether the 146 genes mapped to genome-wide significant loci from our GWAS were enriched for these 83 monogenic ASD genes. We found that 5 of the 146 mapped epilepsy genes were also monogenic ASD genes, which is more than expected by chance (hypergeometric test, $P=4.1 \times 10^{-5}$). This enrichment was slightly less significant when limiting the analysis to our list of 21 biologically prioritized epilepsy genes, which contained 2 ASD genes (hypergeometric test, $P=9.8 \times 10^{-5}$).

Of these 83 ASD genes, there were 11 genes that are established monogenic epilepsy genes, as well as ASD genes. Next, we assessed whether the 146 genes from the current epilepsy GWAS were enriched for these 11 overlapping 'ASD+epilepsy' genes. The list of genes mapped in our epilepsy GWAS contained 2 'ASD+epilepsy' genes (*SCN2A* and *SCN9A*), which is significantly more than expected by chance (hypergeometric test, $P=6.8 \times 10^{-5}$). Similarly, our list of 21 biologically prioritized genes contained one 'ASD+epilepsy' gene (*SCN2A*), which is also more than expected by chance (hypergeometric test, $P=6.2 \times 10^{-5}$).

The largest and most recent ASD GWAS (Grove et al., 2017, BioRxiv), which we also used to compute the genetic correlation between epilepsy and ASD, only found three genome-wide significant loci in their main analysis. These three ASD loci mapped to 29 genes using the same mapping methods applied by us. The 29 ASD genes did not contain a single established monogenic ASD gene, nor any monogenic epilepsy gene. This could suggest that monogenic ASD might be etiologically distinct from the common forms of ASD that were included in the ASD GWAS.

Collectively, these findings suggest the following to us. First, some monogenic genes are clearly involved with both ASD and epilepsy. Second, the lack of a significant genetic correlation between the ASD and the present epilepsy GWAS could be due to a lack in statistical power, but there is unlikely to be a major genetic overlap between the common multifactorial forms. We feel that these observations are outside the scope of the current manuscript and we are including this line of investigation in a further follow-up of our study including more neurological and psychiatric conditions.

I also wonder if there is any relationship to the genetic signal from febrile seizures, where there are a few genome wide hits (PMID: 25344690).

Response 1.5

Phenotypic data on febrile seizures and other comorbidities were not available in our dataset. Collecting such information would require re-phenotyping across thousands of patients many of whom were assessed years ago in adult life. Thus, it was not possible to reliably assess an epidemiological association or phenotypic correlation with co-morbidities *within* our data set.

Finally, the use of the LINCS data to identify drugs was interesting, but not followed up on in any meaningful way. It would be nice to at least see some validation that some of the top drugs, when used in a mouse model, result in the expected gene expression changes. I am sure there are a few drugs out of the top 20 or 30 that have been tested on mice with gene expression data from brain.

Response 1.6

We agree that such data would be of great value, but unfortunately, and to the best of our knowledge, there are no published studies examining the transcriptomic effects of licensed drugs in animal models of epilepsy, and such data is not available for our top drug predictions.

Major issues

-Please clarify upfront which results are from a mega-analysis vs meta-analysis, and whether there is any bona fide replication in this study.

Response 1.7

*We agree that our overall study design was not sufficiently clear. Subjects were assigned to three broad ancestry groups (Caucasian, Asian and African-American) according to results of genotype-based principal component analysis (**Supplementary Figure 1**). We performed pooled linear-mixed model mega-analyses within ancestral groups and epilepsy subtypes, after which trans-ethnic meta-analyses were undertaken. For simplicity, we refer in the manuscript to our GWAS as a mega-analysis to reflect the fact that the majority of our cohort (96%) was analysed in this way. The 'study design' section of the manuscript now states:*

Furthermore, 531 cases of Asian descent, and 147 cases of African descent were included through a meta-analysis. However, we refer to our GWAS as a mega-analysis as the vast majority of our sample (96%) was analysed under that framework.

*In relation to replication, we have also clarified the ‘Study design’ section that now reads:
The current study includes a further 6,516 cases and 3,460 controls in addition to the 8,696 cases and 26,157 controls from our previously published analysis.¹⁵ Thus, this mega-analysis is not a formal replication of our previously published meta-analysis. We do not attempt any formal replication of novel association signals detected by this analysis. Furthermore, 531 cases of Asian descent, and 147 cases of African descent were included through a meta-analysis. However, we refer to our GWAS as a mega-analysis as the vast majority of our samples (96%) were analysed under that framework.*

-Implicating genes with promoters/enhancers and Hi-C and/or eQTL/TWAS data only would be much more convincing than the equal weighting of six arbitrary data types and tallying up scores approach used here

Response – please see Responses 1.1 and 1.2 above

-Please see my comments about genetic correlations and rare variants above. I think it would be worthwhile to: Assess the co-heritability of febrile seizures using the existing GWAS (there is a known epidemiological risk association) and perhaps GWAS for other disorders that are known to be co-morbid with genetic epilepsies

Response – please see Response 1.4 above

--Assess enrichment with MAGMA or stratified LD score regression (PMID 26414678) with genes or the promoters/enhancers of genes harboring rare variants in epilepsy and ASD. These analyses would test biological hypothesis about epilepsy comorbidities and re-assure me that Figure 4 is accurately computed.

Response – please see Response 1.4 above

Minor points

-It would be informative to see whether there is differential enrichment in genes expressed prenatally vs postnatally vs in later life across the epilepsies.

Response – please see Response 1.3 above

-Mouse model validation with existing data for some of the drugs would add excellent support for the otherwise highly speculative connectivity map analysis, as mentioned above.

Response – please see Response 1.6 above

-Figure 2D and S10 don’t have labels on the x-axis. I would cut down the number of bars and just show the top 10, with labels underneath.

Response 1.8

We thank the reviewer for this suggestion and we have now edited section D of each figure to show the top 10 bars in more detail while highlighting the overall graph in an inset image. We feel that the skew towards brain tissue is an important message in this figure and one we want to preserve.

D

Heritability enrichment of 6 different chromatin markers in 88 tissues (Roadmap) in generalized epilepsy

E

Heritability enrichment of gene expression in 53 tissues (GTEx) in generalised epilepsy

Heritability enrichment of 6 different chromatin markers in 88 tissues (Roadmap) in all epilepsy

Heritability enrichment of 6 different chromatin markers in 88 tissues (Roadmap) in focal epilepsy

Heritability enrichment of 6 different chromatin markers in 88 tissues (Roadmap) in generalized epilepsy

-For the enrichment analysis for common variants in monogenic epilepsy genes and antiepileptic drugs: background size, odds ratio, and rationale for background sets would be informative. It is known that longer genes are more highly expressed in brain and this can bias results, etc.

Response 1.9

We thank the reviewer for these suggestions. We have now included in the manuscript, odds ratios for all hypergeometric enrichment tests. As the background set for our hypergeometric testing, we have used all protein coding genes (n=19,180) defined by ANNOVAR. We have used this as a background set since we have used ANNOVAR (as implemented in FUMA) to map the loci of our GWAS to genes. Hence this list of 19,180 genes constitutes all genes that could have been mapped in our GWAS. The average size of all 19,180 genes is 62.7kb. The average size of all our 146 mapped genes is no larger (65.6kb), suggesting there is no strong bias towards preferentially mapping loci to small or large genes. However, as a conservative estimate we have now complemented our hypergeometric tests with a test corrected for gene size, with which we found comparable results as we now describe in an expanded section of the Supplementary methods (see below), and refer to this in the main text, section Enrichment analyses pg 8, line 9: **We did not find a bias for gene size in our enrichment analyses when using a conservative method to correct for this (Supplementary Methods).**

Supplementary methods

Enrichment analyses corrected for gene size

The gene-size of brain expressed genes is known to be longer than non-brain expressed genes. To assess whether gene size could be a cause of bias for our enrichment analyses, we first assessed whether the gene size of the genes mapped in our analyses was different than non-mapped genes in the genome. We found that the size of the 146 genes mapped to genome-wide significant loci was 65.6kb, whereas the average gene size of all other protein-coding genes is on average 62.2kb, suggesting there is no strong bias towards preferentially mapping loci to small or large genes.

We did find that the 102 established monogenic epilepsy genes are on average 2.44 times longer than non-epilepsy genes (152.0kb vs 62.2kb). As a conservative approach to correct for this size difference, we have used the Wallenius' noncentral hypergeometric distribution, as implemented in the R-package 'BiasedUrn'. Using this distribution, we have repeated our hypergeometric analyses under the conservative assumption of a 2.42 times increased likelihood of mapping epilepsy genes as opposed to non-epilepsy genes. Using this distribution we found that the 146 genes that were mapped to genome-wide significant loci were significantly enriched for monogenic epilepsy genes ($p=8.3 \times 10^{-3}$). When limiting our results to the 21 biological prioritized genes we found that the enrichment of monogenic epilepsy genes became more significant ($p=5.3 \times 10^{-4}$).

Similarly, we found that the targets of AEDs are on average 2.43 times longer than non-AED target genes (151.8kb vs 62.4kb). When correcting for this gene-size difference under the

assumption of a 2.43 times increased likelihood of mapping our genome-wide significant loci to AED target genes, we find that the 146 mapped genes were significantly enriched for AED target genes ($p=1.7 \times 10^{-5}$).). When limiting our results to the 21 biological prioritized genes we found that the enrichment of AED target genes became more significant (1.0×10^{-8}).

-It would be helpful to discuss if future studies will be on refractory vs non-refractory epilepsy, or if there might be a way to study drug responsiveness in epilepsies

Response 1.10

We agree with the reviewer that studies on drug responsiveness in relation to epilepsy genetics is an interesting and important avenue of research. The reviewer might be pleased to know that several investigators from our consortium are part of the Epilepsy Pharmacogenetics (EpiPGX) Consortium, which has several ongoing projects to study drug responsiveness and adverse drug reactions based on genetic data.

While these analyses are beyond the scope of the current study, we have now incorporated these prospects in our conclusion pg10, line15: **Future studies including pharmacoresponse data, imaging, and other clinical measurements have the potential to further increase the benefit of these studies for people with epilepsy.**

Overall, this is a very interesting and valuable study that further characterizes the contribution of common variation to different subtypes of epilepsy. The functional analyses with the GWAS data were very extensive, though not coherent and had some results that were counterintuitive and were not well explained. It would be helpful to have additional analyses that support the weaker analyses and some additional discussion about these results.

Response 1.11

We thank the reviewer for the extensive and helpful comments. We believe these issues have been addressed above. The discussion has been expanded to deal with these issues.

Reviewer #2

This is a well conducted and clearly presented investigation into the genetic bases of the epilepsies. They have a large series of patients >15K and almost twice as many controls. They have categorised the patient into one of 3 groups- focal epilepsies, genetic generalised and unclassified. As part of the further analyses they have also investigated the role of the identified loci in some of the well recognised sub-classifications- eg JME, CAE etc.

The samples of been genotyped using widely available arrays and appropriate QC has been applied. The history of this sort of broad approach of using GWAS in the epilepsies was largely little disappointing in the early years- took small or underpowered studies. This despite there being high expectation, due the heritability estimates, that GWAS ought to offer a successful approach. The first chinks appeared a few years ago with the largest scale (at that time) and now this. to this reviewers knowledge the largest, most comprehensive study so far. They report several new loci and confirm 3 of the previously identified loci.

I think it will be important for them to specify when compared to the previous gWAS by this group involving 8696 cases and this one with 15.2K cases if there is any overall;p ie are there 7.5K new cast or are these a completely new set. This will help the reader know how much weight to put by the confirmation of the older loci.

Response 2.1

We have now clarified this in the main text. We have provided a more detailed description in the section 'study design' that now reads: **The current study includes a further 6,516 cases and 3,460 controls in addition to the 8,696 cases and 26,157 controls from our previously published analysis.¹⁵ Thus, this mega-analysis is not a formal replication of our previously published meta-analysis. We do not attempt any formal replication of novel association signals detected by this analysis. Furthermore, 531 cases of Asian descent, and 147 cases of African descent were included through a meta-analysis. However, we refer to our GWAS as a mega-analysis as the vast majority of our samples (96%) were analysed under that framework.**

I also think a little note one caution needs to be struck as these findings have not yet been replicated in a further separately acquired cohort.

I think particular caution in interpretation in the sub-group analyses- ie when a gene is being reported as significant in a clinical sub-type. It would b very reassuring if a second similarly phenotyped cohort could be analysed. I appreciate however that it is very unlikely that this consortium has available another large series of patients but I do think the need to reflect these issues in their paper.

Response 2.2

We agree with the reviewer that this issue needs to be highlighted and have included the following sentence in the discussion (Discussion pg 9, 3rd paragraph), last sentence now reads: **However, the relatively low sample size of our subtype analysis warrants a conservative interpretation and follow-up with a larger cohort.**

The second part of their manuscript deal with a variety of approaches including TWAS etc to try and prioritise the underlying gene as opposed to just describing the loci. I think this does add value and although there are no current absolute 'gold' standards to prove a gene they have used a variety of tools and analyses to bolster the evidence behind the genes named. Whether all of these genes will in time (and replication) prove to be the gene remains to be seen. It is also intriguing to note, as the authors discuss, that a significant proportion of these prioritised genes are known drug targets supporting the view that maybe some of the eaters provide putative targets. Overall I though this was an impressive piece of work.

Response 2.3

Thank you! We agree with these thoughts and believe they are covered in the manuscript.

Reviewer #3

The ILAE consortium performed the largest GWAS on epilepsy, almost doubling the number of cases in their study, and identify 11 novel loci. The manuscript is rich in follow-up analyses, but lacks in depth discussion of the clinical and biological implications of their results. With proper revisions, this manuscript will be well suited for the journal.

Overall:

1. Understandably, the paper is transferred and formatted as a brief communication, but a revised version should make use of the available space and increase the depth and clarity (see other points below).

Response 3.1

We thank the reviewer for noting this. We have now expanded our manuscript and added more clarification throughout. In particular we added text to the introduction and added a discussion section.

Methods:

2. While this analysis increases the number of cases to 15k, one wonders whether the researchers were not able to incorporate additional data from large biobank such as the UKB. Was epilepsy simply not ascertained in any other dataset?

Response 3.2

The UK-biobank obviously represents a valuable resource for GWAS studies. We have thoroughly looked into the data available in the UKB but concluded that the amount of epilepsy cases was relatively small (1,717), while the quality of phenotyping/epilepsy diagnostics was insufficient to be used for our analysis. Although we hope to expand our sample size when new datasets become available, we are unaware of any currently available dataset with detailed epilepsy phenotyping that has not yet been included in our analyses.

3. Hypergeometric test: to what extent were other factors taken into account for the enrichment analyses (e.g., gene size, location)? when comparing to all protein-coding genes, it is not unreasonable to imagine GWAS hits (regulatory variants) in general may enrich specific gene sets. Did the authors also perform this analysis (enrichment for monogenic genes) for GWAS hits of other traits, e.g. height / diabetes / alzheimers?

Response 3.3

Please see our response 1.4 to a similar question from Reviewer 1 which addresses the general issues regarding the hypergeometric test. We have now updated **page 8** of the manuscript and **page 3** of the supplements to address this concern.

Regarding the specific issue of enrichment in other traits, we would like to thank the reviewer for this interesting suggestion. As a negative control, we have now compared our (epilepsy) findings to the publicly available GWAS on Alzheimer’s disease, height, type 2 diabetes and autism spectrum disorder (ASD). We used the same parameters as in our epilepsy GWAS to map genome-wide significant loci to protein-coding genes, after which we performed hypergeometric tests to assess whether these mapped genes were enriched for monogenic epilepsy genes or AED target genes. We did not find any significant enrichment of monogenic epilepsy genes or AED target genes for any of these GWAS (see table below).

GWAS	PMID	# of genome-wide significant loci	# of mapped genes	# of monogenic epilepsy genes	Hypergeometric test for enrichment of monogenic epilepsy genes	# of AED target genes	Hypergeometric test for enrichment of AED target genes
Alzheimer’s disease	24162737	12	168	1	P=0.22	1	P=0.59
Height	25282103	487	5790	19	P=0.99	12	P=0.97
Type 2 Diabetes	28566273	26	179	0	P=0.54	0	P=0.38
Autism spectrum disorder	Grove et al., 2017 (Biorxiv)	3	29	0	P=0.14	0	P=0.09

4. Genetic correlation: did the authors consider neuroimaging traits? hippocampal volume, icv, other subcortical structures.

Response 3.4

We agree that this is an interesting idea to study neuroimaging traits in relation to genetics of epilepsy and such studies are underway through our close collaboration with the ENIGMA neuroimaging consortium. Unfortunately, for the present data set, there are relatively few cases with available high-quality MR images to look at neuroimaging traits directly.

Results:

5. The manhattan plot in figure 1 could be made more informative by marking the novel loci and perhaps gene/snp names.

Response 3.5

We thank the reviewer for this helpful suggestion and we have now updated Figure 1 to distinguish between previously known loci (black) and novel loci (red) from this study.

6. I miss a table/figure/heatmap in the main text showing the associations of all 11(/16) loci with all subtypes. Perhaps this could even replace some of the text in the results section.

Response 3.6

We agree with the reviewer that a figure showing the association of all loci with all subtypes is informative. We had already created such a figure which was placed in the supplements due to space limits (**Supplementary Figure 6**). We have added text to further highlight this analysis, and at the same, emphasize that caution need to be taken given the relatively small sample size of the subtypes. The section “Genome- wide associations”, second paragraph, now reads:

Further analysis of the association signals for each locus in the different syndromes suggested that some signals display specificity for a single subtype, while others show evidence for pleiotropy (**Supplementary Figure 6**). However, the relatively small sample sizes of these phenotype subsets warrants caution for over-interpretation.

Supplementary Figure 6: Forest plot of the lead SNPs from genome-wide significant loci in the 7 epilepsy subtypes. The subtype where the lead SNP was identified is displayed in parentheses, followed by a representative gene in the locus. Beta regression coefficients with standard errors from BOLT-LMM are displayed on the X-axis.

Discussion:

7. Perhaps if/when the authors reformat their paper, a more thorough discussion could be added. What does this major advance mean for the? does it point to subtypes that

could merged, or redefining diagnoses altogether give the lack of heritability? Or are these subtypes real but simply environmentally driven. Could these findings help with diagnosis making, perhaps prognosis/prediction? The biological discussion does go a bit further at times, but it would be nice if the authors can leverage the extra space they now have.

Response 3.7

We agree with the reviewer and have added a more thorough discussion. This sections now reads:

Discussion

The increased sample size in this second ILAE Consortium GWAS of common epilepsies has resulted in the detection of 11 novel risk loci for epilepsy and illustrates how common variants play an important role in the susceptibility of these conditions. But compared to other common neurological diseases our sample size is modest. For example the latest GWAS in schizophrenia considered 36,989 schizophrenia cases and 113,075 controls, resulting in the identification of 108 risk loci.⁵¹ Larger efforts would deliver further insight to the genetic architecture of the common epilepsies.

The majority of the novel loci are associated with GGE. This observation is a welcome partial explanation for the high heritability of GGE, in light of the relative lack of rare variant variants discovered to date. We also show that there is substantial genetic correlation between GGE syndromes. We speculate that the GGE subtypes share a large part of the genetic susceptibility for generalized epilepsies, with specific modifying factors determining the specific syndrome.

Some syndrome-specific associations were detected, such as the relatively strong signal for STX1B in JME, and the association of GJA1 with FE-hippocampal sclerosis. Interestingly, STX1B has been previously linked to a spectrum of epilepsies, in which rare pathogenic variants are believed to cause fever-associated epilepsy syndromes, epileptic encephalopathies and genetic generalized epilepsies, including JME and epilepsy with myoclonic-atonic seizures.^{24,25} Further, mutations in the gap-junction gene GJA1 associated with impaired development of the hippocampus²⁷ and different expression has been reported in epileptic hippocampal and cortical tissue.^{28,29} These findings represent a tantalising glance of the different biological mechanisms underlying epilepsy syndromes that may guide us to the introduction of genetics for improved diagnosis, prognosis and treatment for these common epilepsies. However, the relatively low sample size of our subtype analysis warrants a conservative interpretation and larger follow-up.

At least three association signals are shared between FE and GGE. The clearest overlapping signal remains the 2q24.3 locus, as we reported previously.¹⁵ This association signal is however complex and we demonstrate that the locus consists of at least two independent signals. Our Hi-C chromatin analysis suggests the complexity includes levels of functional association to SCN2A and SCN3A, that are located more distal to the SCN1A locus. Mutations in SCN2A and more recently SCN3A are established monogenic causes of EE that, like SCN1A, cause

dysfunction of the encoded ion-channels, which is believed to disturb the fine balance between neuronal excitation and inhibition. This may involve independent variation that either affect regulation of SCN1A, SCN2A, or SCN3A independently. However, the complex association may also reflect multiple rare risk variations, and large resequencing studies will shed further light on this issue.

The number of association signals we detected and increased power relative to our previous meta analysis¹⁵ allowed us to explore the biological mechanisms behind the observed genetic associations. We show that the signals converge on the dorsolateral prefrontal cortex as the tissue in which most functional effect is observed; this is broadly consistent with the importance of the frontal lobes in generalized epilepsies. Indeed, our analyses of the epigenetic markers H3K27ac and H3K4me1, TWAS, and tissue-specific heritability enrichment analysis all point towards epigenetic regulation of gene expression in the dorsolateral prefrontal cortex as a potential pathophysiological mechanism underlying our epilepsy GWAS findings.

Reviewer #1 (Remarks to the Author):

The authors have responded to my comments in a satisfactory manner. Only one issue remains.

In Response 1.5, the authors state that their cohort did not have information about which patients had febrile seizures. I am not asking for the authors to use their own data to assess the relationship between their findings and febrile seizures, but to perform a genetic correlation with the summary statistics. This should be very doable. I get the impression this is also what reviewer #3 means when the authors reply in Response 3.4 that they cannot compare to neuroimaging traits.

I would recommend the above point be addressed. Otherwise, this is an excellent study that I anticipate will be very interesting to a wide audience.

Reviewer #2 (Remarks to the Author):

They have improved the paper.

They provide greater clarity in the main findings/impact of this study but also cover some of the limitations.

They have made a thorough and thoughtful response to the reviewers concerns

I have no further issues. Think it adds to our understanding of this complex but common disease.

Reviewer #3 (Remarks to the Author):

Nice revision, suitable for publication.

Response to Referee comments for Manuscript NCOMMS-18-15300A

We include the complete text of each reviewer shown in **bold**. Our response to each individual comment is detailed below with specific manuscript text changes in **red**.

Reviewer #1 (Remarks to the Author):

The authors have responded to my comments in a satisfactory manner. Only one issue remains.

In Response 1.5, the authors state that their cohort did not have information about which patients had febrile seizures. I am not asking for the authors to use their own data to assess the relationship between their findings and febrile seizures, but to perform a genetic correlation with the summary statistics. This should very doable. I get the impression this is also what reviewer #3 means when the authors reply in Response 3.4 that they cannot compare to neuroimaging traits.

I would recommend the above point be addressed. Otherwise, this is an excellent study that I anticipate will be very interesting to a wide audience.

We thank Reviewer #1 for their overall positive feedback and apologise for not making our response to item 1.5 clear.

We had indeed performed the suggested experiment to determine whether there was genetic correlation with febrile seizures. We report the findings in Table 4 and Supplementary Table 7, but we did not refer specifically to these results in the text, which was an oversight given the interest of this question. We have now inserted a sentence “Perhaps surprisingly, we did not find significant correlations with febrile seizures.” at the end of the results section that makes this important point clearer (page 8; last paragraph).

New text in red:

*In view of the increasing data on comorbidities with epilepsy, we next used LD-score regression to analyse the genetic correlation between epilepsy and various other brain diseases and traits from previously published GWAS (Figure 4; see Supplementary Table 7 for values). **Perhaps surprisingly, we did not find significant correlations with febrile seizures. Similarly, we did not find any significant genetic correlations between epilepsy and other neurological or psychiatric diseases.***

Reviewer #2 (Remarks to the Author):

They have improved the paper.

They provide greater clarity in the main findings/impact of this study but also cover some of the limitations.

They have made a thorough and thoughtful response to the reviewers concerns

I have no further issues. Think it adds to our understanding of this complex but common disease.

We thank Reviewer #2 for their positive feedback.

Reviewer #3 (Remarks to the Author):

Nice revision, suitable for publication.

We thank Reviewer #3 for their positive feedback.

Reviewer #1 (Remarks to the Author):

The authors have addressed all of my concerns. It is a very interesting and valuable study!

REVIEWERS' COMMENTS:

Reviewer #1 (Remarks to the Author):

The authors have addressed all of my concerns. It is a very interesting and valuable study!

Response: We thank Reviewer #1 for their positive comments